# Cntnap2 loss drives striatal neuron hyperexcitability and behavioral inflexibility

Katherine R Cording[1,2], Emilie M Tu[1,3], Hongli Wang[2], Alexander HCW Agopyan-Miu[3], Helen S Bateup[1,2,3]*

[1]Helen Wills Neuroscience Institute, University of California, Berkeley, Berkeley, United States; [2]Department of Neuroscience, University of California, Berkeley, Berkeley, United States; [3]Department of Molecular and Cell Biology, University of California, Berkeley, Berkeley, United States

## eLife Assessment

This **important** and well-executed study describes how deleting the autism spectrum disorder risk gene CNTNAP2 in mice increases dorsolateral striatal projection neuron excitability and promotes repetitive behaviors and cognitive inflexibility. The evidence supporting this claim is **convincing**. The study provides a potential cellular explanation for the repetitive and inflexible behavior in Cntnap2 knockout mice and CNTNAP2 disorder in humans, which would interest both basic and translational neuroscientists.

**\*For correspondence:**
bateup@berkeley.edu

**Competing interest:** The authors declare that no competing interests exist.

**Abstract** Autism spectrum disorder (ASD) is a neurodevelopmental disorder characterized by two major diagnostic criteria – persistent deficits in social communication and interaction, and the presence of restricted, repetitive patterns of behavior (RRBs). Evidence from both human and animal model studies of ASD suggests that alteration of striatal circuits, which mediate motor learning, action selection, and habit formation, may contribute to the manifestation of RRBs. *CNTNAP2* is a syndromic ASD risk gene, and loss of function of *Cntnap2* in mice is associated with RRBs. How the loss of *Cntnap2* impacts striatal neuron function is largely unknown. In this study, we utilized *Cntnap2*[-/-] mice to test whether altered striatal neuron activity contributes to aberrant motor behaviors relevant to ASD. We find that *Cntnap2*[-/-] mice exhibit enhanced cortical drive of direct pathway striatal projection neurons (dSPNs). This enhanced drive is due to increased intrinsic excitability of dSPNs, which make them more responsive to cortical inputs. We find that *Cntnap2*[-/-] mice exhibit spontaneous repetitive behaviors, increased motor routine learning, perseveration, and cognitive inflexibility. Increased corticostriatal drive may therefore contribute to the acquisition of repetitive, inflexible behaviors in *Cntnap2* mice.

## Introduction

Autism spectrum disorder (ASD) is characterized by alterations in social communication and interaction, as well as the presence of restricted, repetitive, inflexible behaviors (*APA, 2022*). Given that ASD has high heritability (*Sandin et al., 2017*), much work has been done in the last 30 years to identify genes that confer risk of developing ASD (*De Rubeis et al., 2014*; *Iossifov et al., 2014*; *Sanders et al., 2015*). Through this, hundreds of high-confidence risk genes have been identified, varying greatly in the proteins for which they code (*Satterstrom et al., 2020*). These include transcriptional and translational regulators, ion channels, receptors, cell adhesion molecules, and others (*De Rubeis*

*et al., 2014; Delorme et al., 2013; Ebert and Greenberg, 2013*). Recent work has focused on identifying brain regions and circuits that may be commonly affected by ASD-related mutations. The basal ganglia, in particular the striatum, represents one such commonly altered brain region, and prior studies have demonstrated changes in striatal function and striatum-associated behaviors in mice with mutations in ASD risk genes (*Benthall et al., 2021; Fuccillo, 2016; Peça et al., 2011; Peixoto et al., 2016; Platt et al., 2017; Rothwell et al., 2014; Wang et al., 2016*). However, whether basal ganglia circuits are convergently changed in ASD mouse models is an open question. Here, we investigated whether loss of function of the syndromic ASD risk gene *Cntnap2* alters striatal physiology and basal-ganglia-dependent behaviors in mice.

*Cntnap2* codes for a neurexin-like cell adhesion molecule called Contactin-associated protein-like 2 (Caspr2; *Poliak et al., 1999; Poliak et al., 2003*). In mice, Caspr2 is expressed in several cortical and subcortical regions, including the striatum, from embryonic day 14 (E14) onward into adulthood (*Peñagarikano et al., 2011*). Caspr2 is primarily localized at the juxtaparanodes of myelinated axons and is involved in the clustering of potassium channels (*Poliak et al., 2003; Scott et al., 2019*). In vitro studies in mice suggest that Caspr2 may also play a role in AMPAR trafficking and cell morphology (*Anderson et al., 2012; Gdalyahu et al., 2015; Varea et al., 2015*), and ex vivo experiments indicate that it can control cell excitability and circuit synchronicity (*Martín-de-Saavedra et al., 2022*). Caspr2 is important during neurodevelopment and has been implicated in neuronal migration (*Peñagarikano et al., 2011*), the maturation and function of parvalbumin-positive GABAergic interneurons (*Peñagarikano et al., 2011; Scott et al., 2019; Vogt et al., 2018*), and the timing of myelination (*Scott et al., 2019*). *CNTNAP2* mutations in people lead to a neurodevelopmental syndrome that can include language disorders, epilepsy, obsessive-compulsive disorder, and ASD (*Peñagarikano and Geschwind, 2012; Rodenas-Cuadrado et al., 2014*). A mouse model of this syndrome, *Cntnap2*$^{-/-}$, has been shown to exhibit good face validity for ASD-relevant social and motor behavior alterations (*Brunner et al., 2015; Dawes et al., 2018; Peñagarikano et al., 2011; Scott et al., 2019*). However, the impact of *Cntnap2* loss on striatal physiology and corticostriatal-dependent behaviors has not been comprehensively assessed.

The striatum is primarily composed of GABAergic striatal projection neurons (SPNs), which make up two functionally distinct output pathways: the D1-receptor expressing cells of the direct pathway (dSPNs), which project to substantia nigra pars reticulata (SNr), and the D2-receptor expressing cells of the indirect pathway (iSPNs), which project to external globus pallidus (GPe; *Calabresi et al., 2014; Gerfen and Surmeier, 2011; Kravitz et al., 2010; Tai et al., 2012*). The two types of SPNs are intermixed throughout the striatum and receive excitatory glutamatergic inputs from cortex and thalamus, as well as dopaminergic input from the midbrain (*Ding et al., 2008; Doig et al., 2010; Gerfen and Surmeier, 2011*). Coordinated activity between the populations of SPNs in response to these inputs mediates action selection, motor learning, and habit formation (*Hawes et al., 2015; Santos et al., 2015; Yin et al., 2005; Yin et al., 2006*). Although SPNs comprise upwards of 95% of the cells in the striatum, there are distinct types of GABAergic interneurons that contribute significantly to the inhibitory circuitry of the striatum. Parvalbumin (PV) interneurons, which make up ~2% of the cells in the striatum, provide the largest feedforward inhibition onto SPNs (*Burke et al., 2017*). Changes in the number and/or function of PV interneurons have been identified in several ASD mouse models, including *Cntnap2*$^{-/-}$ mice, implicating PV circuitry as a potential common alteration across ASD mouse models (*Filice et al., 2020; Juarez and Martínez Cerdeño, 2022*).

To determine how the loss of *Cntnap2* affects striatal function, we assessed the physiology of SPNs and PV-interneurons in the dorsolateral striatum (DLS) and utilized a range of assays to assess striatum-associated behaviors in *Cntnap2*$^{-/-}$ mice. We find that SPNs of the direct pathway exhibit increased corticostriatal drive, despite unchanged excitatory cortical synaptic input. Although decreased inhibitory function has been identified in other brain regions in *Cntnap2*$^{-/-}$ mice, we find no deficit in broad or PV-specific inhibitory input onto either SPN subtype in the case of *Cntnap2* loss. Instead, we identify a significant increase in the intrinsic excitability of dSPNs in *Cntnap2*$^{-/-}$ mice, driven by a reduction in Kv1.2 channel function. Behaviorally, we find that *Cntnap2*$^{-/-}$ mice exhibit RRB-like behaviors including increased self-grooming, marble burying, and nose poking in the holeboard assay. *Cntnap2*$^{-/-}$ mice also exhibit increased motor routine learning in the accelerating rotarod and cognitive inflexibility in an odor-based reversal learning task. Taken together, these findings suggest that enhanced direct

pathway excitability may play a role in the spontaneous and learned repetitive behaviors exhibited by *Cntnap2-/-* mice.

## Results

### Cntnap2$^{-/-}$ dSPNs exhibit increased cortical drive

Emerging evidence indicates that corticostriatal synapses are a common site of alteration in mouse models of ASD (*Li and Pozzo-Miller, 2020*). To test whether mice with loss of *Cntnap2* exhibit changes in corticostriatal connectivity, we crossed *Cntnap2$^{-/-}$* mice to *Thy1*-ChR2-YFP mice, which express channelrhodopsin in a subset of layer V pyramidal neurons (*Figure 1A*; *Arenkiel et al., 2007*; *Poliak et al., 1999*). These mice were crossed to a D1-tdTomato reporter line to visually identify dSPNs (*Ade et al., 2011*). We recorded from SPNs in the DLS, as this sensorimotor striatal subregion is implicated in the acquisition of habitual and procedural behaviors (*Packard and Knowlton, 2002*). Changes in physiological function in this area may be connected to the acquisition of repetitive motor behaviors in ASD (*APA, 2022*; *Fuccillo, 2016*). To simulate a train of cortical inputs, we applied ten pulses of blue light over the recording site in DLS and measured the number of action potentials (APs) fired by SPNs in the absence of synaptic blockers (*Figure 1A*). We altered the light intensity to vary the probability of eliciting subthreshold depolarizations or AP firing. dSPNs were identified using tdTomato fluorescence, and tdTomato-negative neurons were designated putative iSPNs.

We quantified the number of evoked APs at different light intensities and found that dSPNs in young adult *Cntnap2$^{-/-}$* mice exhibited increased spike probability compared to wild-type (WT) dSPNs (*Figure 1B and C*). The interaction effect of genotype and stimulation intensity in these cells suggests increased corticostriatal drive, consistent with findings in another mouse model with loss of function of the ASD-risk gene *Tsc1* (*Benthall et al., 2021*). *Cntnap2$^{-/-}$* iSPNs had subtly increased cortically-evoked AP firing compared to WT iSPNs, although this was not statistically significant (*Figure 1D and E*). To test whether the enhanced spiking probability of *Cntnap2$^{-/-}$* dSPNs was due to excitatory synaptic changes, we applied blue light of varying intensity over the recording site in DLS while holding cells at −70 mV to evoke AMPAR-driven excitatory postsynaptic currents (EPSCs). We found that the average optically evoked EPSC amplitude was not significantly different across a range of light intensities in *Cntnap2$^{-/-}$* dSPNs or iSPNs compared to WT (*Figure 1F–I*). In an additional group of mice, we measured the ratio of AMPAR currents recorded at −70 mV to NMDAR currents recorded at +40 mV (at 20% blue light intensity) and found no significant differences in AMPA:NMDA ratio in *Cntnap2$^{-/-}$* dSPNs or iSPNs (*Figure 1J–M*).

To further assess synaptic inputs, we measured the number of dendritic spines in *Cntnap2$^{-/-}$* and WT SPNs, which are typically the sites of cortical synaptic innervation (*Bouyer et al., 1984*; *Xu et al., 1989*). To visualize spines, we injected neonate *Cntnap2;Drd1a*-tdTomato mice with AAV5-*Syn1*-GFP virus to sparsely label dSPNs and iSPNs in the DLS (*Figure 1—figure supplement 1*; *Keaveney et al., 2018*). We found that *Cntnap2$^{-/-}$* SPNs in adult mice had similar spine density as WT (*Figure 1—figure supplement 1*), suggesting no overall change in synapse number. Together, these results show that dSPNs in *Cntnap2$^{-/-}$* mice exhibit enhanced cortically driven spiking. However, this is not due to a change in corticostriatal synaptic strength or overall synapse density.

### Inhibition is unchanged in Cntnap2$^{-/-}$ SPNs

Previous work has indicated a reduction in the number and/or function of fast-spiking parvalbumin-expressing (PV) interneurons across multiple brain regions in *Cntnap2$^{-/-}$* mice (*Ahmed et al., 2023*; *Antoine et al., 2019*; *Jurgensen and Castillo, 2015*; *Paterno et al., 2021*; *Peñagarikano et al., 2011*; *Vogt et al., 2018*). While deficits in inhibition have been identified in the cortex and hippocampus (*Antoine et al., 2019*; *Jurgensen and Castillo, 2015*), and the number of PV interneurons has been reported to be decreased in the striatum (*Peñagarikano et al., 2011*), a comprehensive assessment of striatal inhibitory synaptic function has yet to be completed in *Cntnap2$^{-/-}$* mice. We first determined whether there were global deficits in inhibition onto SPNs in *Cntnap2$^{-/-}$* mice using intra-striatal electrical stimulation to evoke inhibitory postsynaptic currents (IPSCs; *Figure 2A*). In *Cntnap2$^{-/-}$* dSPNs and iSPNs, the average amplitude of electrically evoked IPSCs across a range of stimulation intensities was not different from WT (*Figure 2B–E*).

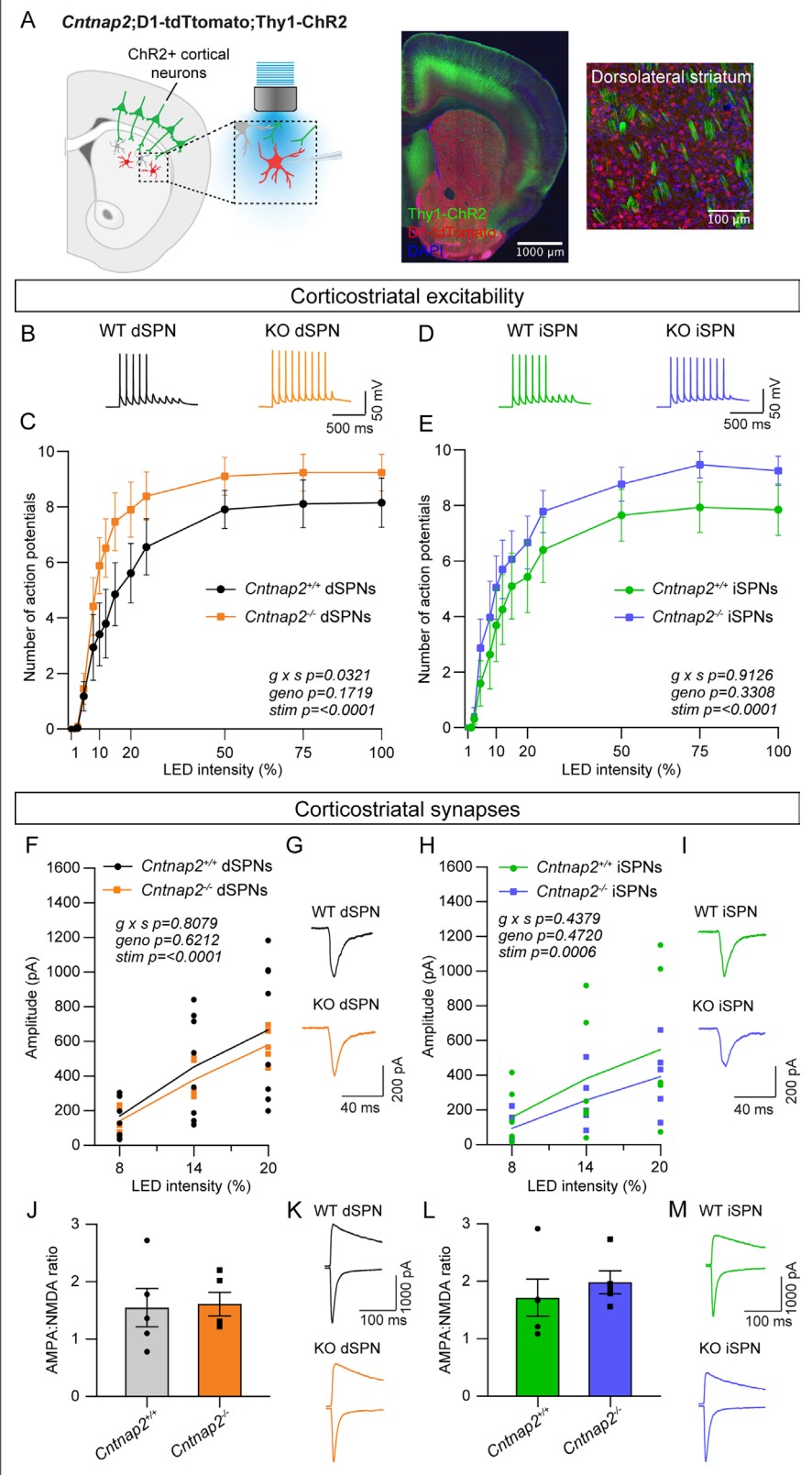

**Figure 1.** *Cntnap2*<sup>-/-</sup> dSPNs exhibit increased cortical drive. (**A**) Left: schematic of the corticostriatal connectivity experiments. For corticostriatal excitability, cortical terminals expressing ChR2 were stimulated with 10 pulses of blue light at 10 Hz and responses were recorded from dSPNs (red) and iSPNs (grey) in dorsolateral striatum. For corticostriatal synaptic strength, cortical terminals expressing ChR2 were stimulated with blue light at increasing

*Figure 1 continued on next page*

*Figure 1 continued*

intensity and synaptic currents were recorded from dSPNs (red) and iSPNs (grey) in dorsolateral striatum. Center: 10 x confocal image of the striatum from a *Cntnap2*[+/+];D1-tdTomato;Thy1-ChR2 mouse. Right: 20 x confocal image of dorsolateral striatum from a *Cntnap2*[+/+];D1-tdTomato;Thy1-ChR2 mouse. YFP (green) labels cell bodies and axons of a subset of layer V pyramidal neurons, tdTomato (red) labels dSPNs, and DAPI stained nuclei are in blue. (**B**) Example single traces of action potentials (APs) in dSPNs evoked by cortical terminal stimulation at 20% light intensity for the indicated genotypes. (**C**) Quantification (mean ± SEM) of the number of APs evoked in dSPNs at different light intensities. *Cntnap2*[+/+] n = 9 mice, 24 cells, *Cntnap2*[-/-] n=10 mice, 22 cells. Repeated measures two-way ANOVA p values are shown; g x s $F_{(12, 204)}$=1.935, geno $F_{(1, 17)}$=2.034, stim $F_{(1.931, 32.83)}$=86.12. (**D**) Example single traces of APs in iSPNs evoked by cortical terminal stimulation at 20% light intensity for the indicated genotypes. (**E**) Quantification (mean ± SEM) of the number of APs evoked in iSPNs at different light intensities. *Cntnap2*[+/+] n = 9 mice, 23 cells, *Cntnap2*[-/-] n=10 mice, 21 cells. Repeated measures two-way ANOVA p values are shown; g x s $F_{(12, 216)}$=0.5012, geno $F_{(1, 18)}$=0.9989, stim $F_{(2.331, 41.96)}$=60.62. (**F**) Average EPSC traces from example dSPNs of each genotype induced by optogenetic cortical terminal stimulation at 14% light intensity. (**G**) Quantification of EPSC amplitude evoked in dSPNs at different light intensities (line represents the mean, dots/squares represent average EPSC amplitude for each mouse). *Cntnap2*[+/+] n = 8 mice, 17 cells, *Cntnap2*[-/-] n=5 mice, 13 cells. Repeated measures two-way ANOVA p values are shown; g x s $F_{(2, 22)}$=0.2154, geno $F_{(1, 11)}$=0.2585, stim $F_{(1.053, 11.58)}$=49.68. (**H**) Average EPSC traces from example iSPNs of each genotype induced by optogenetic cortical terminal stimulation at 14% light intensity. (**I**) Quantification of EPSC amplitude evoked in iSPNs at different light intensities (line represents mean, dots/squares represent average EPSC amplitude for each mouse). *Cntnap2*[+/+] n = 6 mice, 13 cells, *Cntnap2*[-/-] n=5 mice, 11 cells. Repeated measures two-way ANOVA p values are shown; g x s $F_{(2, 18)}$=0.4428, geno $F_{(1, 9)}$=0.5635, stim $F_{(1.095, 9.851)}$=23.82. (**J**) Quantification (mean ± SEM) of AMPA:NMDA ratio in dSPNs evoked by 20% light intensity (dots/squares represent average AMPA:NMDA ratio for each mouse). *Cntnap2*[+/+] n = 5 mice, 22 cells, *Cntnap2*[-/-] n=5 mice, 22 cells, p=0.8413, Mann-Whitney test. (**K**) Example traces show pairs of EPSCs evoked by optogenetic corticostriatal stimulation (20% light intensity) recorded at +40 mV (top traces) and −70 mV (bottom traces) from *Cntnap2*[+/+] and *Cntnap2*[-/-] dSPNs. (**L**) Quantification (mean ± SEM) of AMPA:NMDA ratio in iSPNs evoked by 20% light intensity (dots/squares represent average AMPA:NMDA ratio for each mouse). *Cntnap2*[+/+] n = 5 mice, 21 cells, *Cntnap2*[-/-] n=5 mice, 21 cells, p=0.3095, Mann-Whitney test. (**M**) Example traces show pairs of EPSCs evoked by optogenetic corticostriatal stimulation (20% light intensity) recorded at +40 mV (top traces) and −70 mV (bottom traces) from *Cntnap2*[+/+] and *Cntnap2*[-/-] iSPNs.

The online version of this article includes the following figure supplement(s) for figure 1:

**Figure supplement 1.** *Cntnap2*[-/-] SPNs do not have altered spine density.

There are many sources of inhibition in the striatum (*Burke et al., 2017*), which can all be activated with electrical stimulation. To assess whether inhibition from PV interneurons specifically is altered in *Cntnap2*[-/-] mice, we crossed *Cntnap2*[-/-];D1-tdTomato mice to *Pvalb*-Cre; RCL-ChR2-H134R-EYFP (Ai32) mice to express channelrhodopsin in PV interneurons (*Figure 2F*; *Hippenmeyer et al., 2005*; *Madisen et al., 2010*). We applied a blue light pulse of varying intensity over the recording site to evoke PV interneuron-specific IPSCs in SPNs, in the presence of excitatory synaptic blockers (*Figure 2F*). We found that the average amplitude of optically-evoked IPSCs did not differ significantly in *Cntnap2*[-/-] dSPNs or iSPNs compared to WT controls (*Figure 2G–J*).

To directly measure PV neuron function, we assessed the intrinsic excitability of PV interneurons in *Cntnap2*[-/-] mice. To visualize PV interneurons for recordings, we crossed *Cntnap2*[-/-] mice to *Pvalb*-Cre;RCL-tdT (Ai9) mice (*Figure 2—figure supplement 1A*). Plotting the number of APs fired as a function of current step amplitude indicated that there were no significant differences in the intrinsic excitability of PV interneurons in *Cntnap2*[-/-] mice compared to controls (*Figure 2—figure supplement 1B and C*). There were also no changes in intrinsic cell properties such as rheobase, membrane resistance, capacitance, resting membrane potential, or AP shape in *Cntnap2*[-/-] PV interneurons (*Figure 2—figure supplement 1D–K*).

Given prior reports of altered PV cell number in *Cntnap2*[-/-] mice (*Paterno et al., 2021*; *Peñagarikano et al., 2011*; *Vogt et al., 2018*), we counted PV-expressing cells in the striatum, using immunohistochemistry and fluorescent in situ hybridization. We found no significant difference in the number of PV-positive cells in the dorsal striatum of *Cntnap2*[-/-] mice compared to WT (*Figure 2—figure supplement 2A–F and K–N*). We also found no changes in the average single cell or total level of PV protein expression in the dorsal striatum (*Figure 2—figure supplement 2G–J*). Overall, we did not

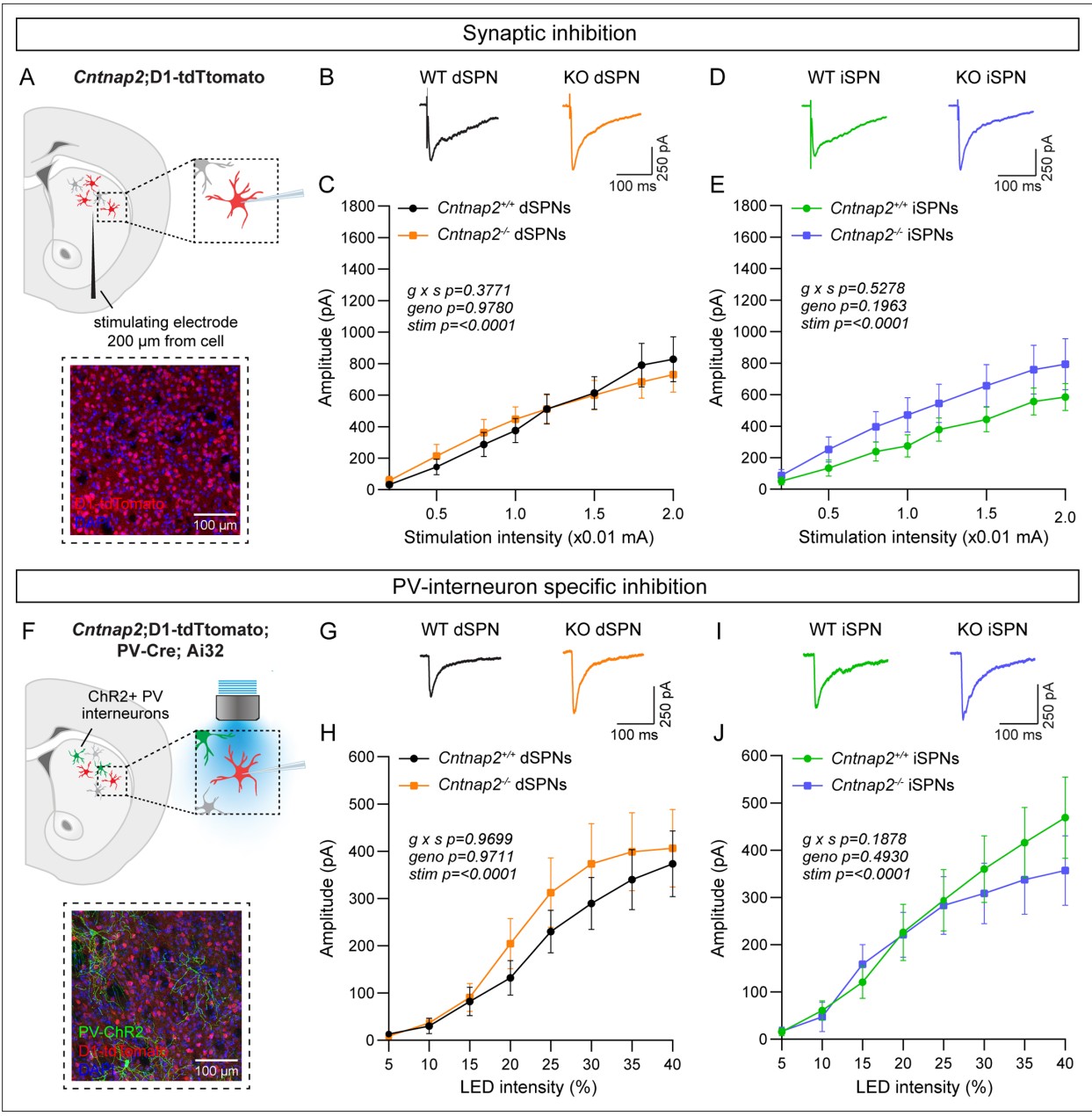

**Figure 2.** Inhibition is not altered in *Cntnap2^-/-* SPNs. (**A**) Top: schematic of the experiment. A bipolar stimulating electrode was placed approximately 200 μm from the recording site. A range of electrical stimulation intensities was applied to the tissue while IPSCs were recorded from dSPNs (red) and iSPNs (grey) in dorsolateral striatum. Bottom: 20 x confocal image of dorsolateral striatum from a *Cntnap2^+/+*;D1-tdTomato mouse. tdTomato (red) labels dSPNs, DAPI stained nuclei are in blue. (**B**) Average IPSC traces from example dSPNs of each genotype evoked by electrical stimulation at 1.5 (x0.01 mA) intensity for the indicated genotypes. (**C**) Quantification (mean ± SEM) of IPSC amplitude in dSPNs at different stimulation intensities. *Cntnap2^+/+* n = 17 cells from 9 mice, *Cntnap2^-/-* n=16 cells from 9 mice. Repeated measures two-way ANOVA p values are shown; g x s F (7, 217)=1.080, geno F (1, 31)=0.0007751, stim F (1.815, 56.28)=54.92. (**D**) Average IPSC traces from example iSPNs of each genotype evoked by electrical stimulation at 1.5 (x0.01 mA) intensity for the indicated genotypes. (**E**) Quantification (mean ± SEM) of IPSC amplitude in iSPNs at different stimulation intensities. *Cntnap2^+/+* n = 16 cells from 9 mice, *Cntnap2^-/-* n=16 cells from 10 mice. Repeated measures two-way ANOVA p values are shown; g x s F (7, 210)=0.8741, geno F (1, 30)=1.746, stim F (1.591, 47.73)=45.66. (**F**) Top: schematic of the experiment. PV interneuron terminals expressing ChR2 were stimulated with blue light at a range of intensities, and optically evoked IPSCs were recorded from dSPNs (red) and iSPNs (grey) in dorsolateral striatum. Bottom: 20 x confocal image of dorsolateral striatum from a *Cntnap2^+/+*;D1-tdTomato;PV-Cre;Ai32 mouse. YFP (green) labels PV interneurons, tdTomato (red) labels dSPNs, DAPI-stained nuclei are in blue. (**G**) Average IPSC traces from example dSPNs of each genotype evoked by optogenetic PV interneuron stimulation at 30% light intensity. (**H**) Quantification (mean ± SEM) of IPSC amplitude in dSPNs at different light intensities. *Cntnap2^+/+* n = 29 cells from 15 mice, *Cntnap2^-/-* n=23 cells from 11 mice. Repeated measures two-way ANOVA p values are shown; g x s F (7, 441)=0.2566, geno F (1, 63)=0.001322, stim F (1.433, 90.25)=32.57. (**I**) Average IPSC traces from example iSPNs of each genotype evoked by optogenetic PV interneuron stimulation at 30% light intensity.

*Figure 2 continued on next page*

*Figure 2 continued*

(J) Quantification (mean ± SEM) of IPSC amplitude in iSPNs at different light intensities. *Cntnap2+/+* n = 24 cells from 14 mice, *Cntnap2-/-* n=27 cells from 13 mice. Repeated measures two-way ANOVA p values are shown; g x s F (7, 343)=1.441, geno F (1, 49)=0.4771, stim F (1.622, 79.46)=38.49.

The online version of this article includes the following source data and figure supplement(s) for figure 2:

**Figure supplement 1.** PV interneuron intrinsic excitability is unchanged in *Cntnap2-/-* mice.

**Figure supplement 2.** *Cntnap2-/-* mice do not exhibit changes in PV-positive cell number or expression.

**Figure supplement 2—source data 1.** PDF file containing original western blots for *Figure 2—figure supplement 2*, indicating the relevant bands and treatments.

**Figure supplement 2—source data 2.** Original files for western blot analysis displayed in *Figure 2—figure supplement 2*.

observe significant changes in PV interneuron number, PV expression, or PV interneuron-mediated inhibition in the adult *Cntnap2-/-* striatum compared to WT controls.

## dSPN intrinsic excitability is increased in Cntnap2-/- mice

Given that the increased cortical drive onto *Cntnap2-/-* dSPNs could not be explained by changes in excitatory or inhibitory synaptic function, we tested whether it could be due to a change in intrinsic excitability. To measure this, we recorded from dSPNs and iSPNs in *Cntnap2;Drd1a*-tdTomato mice and injected current steps of increasing amplitude. We found that *Cntnap2-/-* dSPNs had significantly increased intrinsic excitability compared to WT dSPNs (*Figure 3A–B*). *Cntnap2-/-* dSPNs also had reduced rheobase current (*Figure 3C*), the minimum current required to evoke an AP, as well as increased membrane resistance (*Figure 3D*). While there was a trend towards increased excitability in *Cntnap2-/-* iSPNs, this effect was not statistically significant (*Figure 3G–H*), and these cells did not exhibit changes in rheobase current (*Figure 3I*) or membrane resistance (*Figure 3J*). Membrane capacitance (*Figure 3E and K*), resting membrane potential (*Figure 3F and L*), and AP shape (*Figure 3—figure supplement 1*) were not significantly changed in *Cntnap2-/-* SPNs, although latency to first spike was reduced in both *Cntnap2-/-* dSPNs and iSPNs (*Figure 3—figure supplement 1C and I*). Given the lack of synaptic changes observed in *Cntnap2-/-* dSPNs, the increase in dSPN intrinsic excitability likely underlies their enhanced corticostriatal drive (see *Figure 1*).

## The effects of Kv1.2 blockade are occluded in Cntnap2-/- dSPNs

Caspr2 is known to be involved in the clustering of voltage-gated potassium channels (*Inda et al., 2006*; *Poliak et al., 1999*; *Poliak et al., 2003*), particularly Kv1.2 channels (*Scott et al., 2019*). These channels play an important role in regulating the intrinsic excitability of SPNs (*Nisenbaum et al., 1994*). Blockade of these channels with the drug α-Dendrotoxin (α-DTX) results in increased AP frequency, decreased rheobase current, decreased first AP latency, and decreased AP threshold (*Shen et al., 2004*), particularly in dSPNs (*Lahiri and Bevan, 2020*). Given that *Cntnap2-/-* dSPNs exhibited increased AP frequency (*Figure 3B*), decreased rheobase current (*Figure 3C*), decreased first AP latency (*Figure 3—figure supplement 1C*), and a trend towards decreased AP threshold (p=0.0516; *Figure 3—figure supplement 1B*), we hypothesized that loss of function of Kv1.2 channels could be the mechanism. To test this, we recorded from dSPNs and iSPNs in *Cntnap2;Drd1a*-tdTomato mice and injected current steps of increasing amplitude in the absence or presence of α-DTX (100 nM). We found that α-DTX enhanced AP firing and decreased AP threshold in WT dSPNs (*Figure 4A–C*), but not in *Cntnap2-/-* dSPNs (*Figure 4D–F*). AP width was also significantly increased by α-DTX in WT but not *Cntnap2-/-* dSPNs (*Figure 4—figure supplement 1H and Q*), while AP fast afterhyperpolarization and AP latency were significantly decreased in both WT and *Cntnap2-/-* dSPNs (*Figure 4—figure supplement 1E, I, N and R*). Together, these data show that the effects of α-DTX on *Cntnap2-/-* dSPNs are largely occluded, indicating that Kv1.2 channels are basally altered by loss of Cntnap2. This likely accounts for the enhanced intrinsic excitability of *Cntnap2-/-* dSPNs.

We tested the effects of α-DTX on WT and *Cntnap2-/-* iSPNs and found minimal changes (*Figure 4G–K*, *Figure 4—figure supplement 2*), with no alterations in AP threshold (*Figure 4I and L*) and only a small shift in the input-output relationship of *Cntnap2-/-* iSPNs (*Figure 4K*). The effects of α-DTX on iSPN excitability have been less well documented, and our data suggest a difference in the sensitivity of dSPNs and iSPNs to Kv1 blockade.

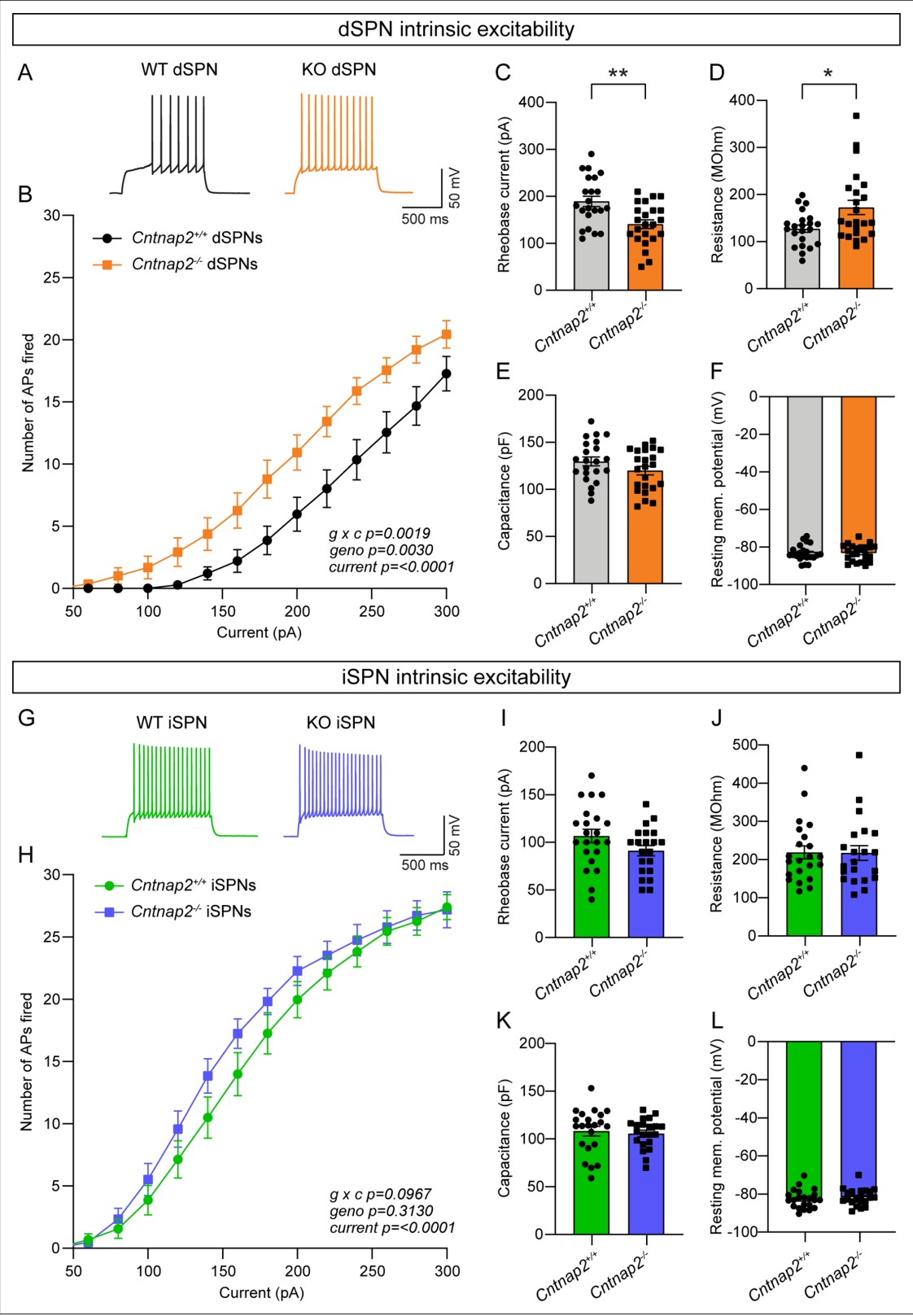

**Figure 3.** Intrinsic excitability is increased in *Cntnap2^-/-^* dSPNs. (**A**) Example AP traces in dSPNs evoked by a 200 pA current step for the indicated genotypes. (**B**) Quantification (mean ± SEM) of the number of APs evoked in dSPNs at different current step amplitudes. *Cntnap2^+/+^* n = 22 cells from 8 mice, *Cntnap2^-/-^* n=23 cells from 8 mice. Repeated measures two-way ANOVA p values are shown; g x c $F_{(12, 528)}$=2.649, geno $F_{(1, 44)}$=107.5, current $F_{(1.974, 86.86)}$=147.5. (**C**) Quantification (mean ± SEM) of the rheobase current in dSPNs. Dots/squares represent the rheobase current for each neuron.

*Figure 3 continued on next page*

*Figure 3 continued*

n is the same as in panel B. **p=0.0016, two-tailed unpaired t test. (**D–F**) Quantification (mean ± SEM) of dSPN membrane resistance (**D**), *p=0.0328, Mann-Whitney test; membrane capacitance (**E**), p=0.2182, Mann-Whitney test; and resting membrane potential (**F**), p=0.9914, two-tailed unpaired t test. Dots/squares represent the average value for each neuron. n is the same as in panel B. (**G**) Example AP traces in iSPNs evoked by a 200 pA current step for the indicated genotypes. (**H**) Quantification (mean ± SEM) of the number of APs evoked in iSPNs at different current step amplitudes. *Cntnap2*$^{+/+}$ n = 22 cells from 8 mice, *Cntnap2*$^{-/-}$ n=21 cells from 8 mice. Repeated measures two-way ANOVA p values are shown; g x c F (12, 516)=1.569, geno F (1, 43)=1.042, current F (2.041, 87.78)=284.7. (**I**) Quantification (mean ± SEM) of the rheobase current in iSPNs. Dots/squares represent the rheobase current for each neuron. n is the same as in panel H. p=0.0923, two-tailed unpaired t test. (**J–L**) Quantification (mean ± SEM) of iSPN membrane resistance (**J**), p=0.8193, Mann-Whitney test; membrane capacitance (**K**), p=0.6886, two-tailed unpaired t test; and resting membrane potential (**L**), P=0.4859, two-tailed unpaired t test. Dots/squares represent the average value for each neuron. n is the same as in panel H.

The online version of this article includes the following figure supplement(s) for figure 3:

**Figure supplement 1.** Latency to spike is reduced in *Cntnap2*$^{-/-}$ SPNs.

## Cntnap2$^{-/-}$ mice display increased repetitive behaviors

RRBs comprise one of the primary symptom domains of ASD (**APA, 2022**). Alterations in striatal circuits are thought to be involved in the manifestation of RRBs, given the striatum's role in action selection and motor control (**Estes et al., 2011**; **Fuccillo, 2016**; **Hollander et al., 2005**; **Langen et al., 2014**). To determine whether changes in motor behavior accompanied the altered striatal physiology in *Cntnap2*$^{-/-}$ mice, we assessed locomotor activity and spontaneous repetitive behaviors using the open field, marble burying, and holeboard assays (**Figure 5A–D**). In the open field, we found no significant difference in the total distance traveled, average speed, or number of rears in *Cntnap2*$^{-/-}$ mice compared to WT controls (**Figure 5E–G**). We did find that *Cntnap2*$^{-/-}$ mice made significantly more entries into the center of the open field arena than WT mice, which may reflect a reduction in avoidance behavior in these mice (**Figure 5H**). Manually scored grooming behavior revealed that *Cntnap2*$^{-/-}$ mice initiated more self-grooming bouts than WT controls in the open field (**Figure 5I**), consistent with prior reports (**Peñagarikano et al., 2011**).

To further assess motor behaviors in *Cntnap2*$^{-/-}$ mice, we utilized the marble burying assay (**Figure 5C**), which takes advantage of a mouse's natural tendency to dig or bury. The number of marbles buried is used as a measure of persistent or repetitive behavior (**Angoa-Pérez et al., 2013**). We found that *Cntnap2*$^{-/-}$ mice buried significantly more marbles on average than WT controls (**Figure 5J**). Another measure of repetitive behavior, which is based on the natural exploratory behavior of mice, is the holeboard assay (**Figure 5D**). In this test, the number of nose pokes made into unbaited holes is recorded. *Cntnap2*$^{-/-}$ mice made significantly more nose pokes within a 10-min period than WT mice (**Figure 5K**). This was largely due to increased poking during the last 5 min of the test (**Figure 5L and M**), indicating persistent poking behavior in *Cntnap2*$^{-/-}$ mice. Together, the increased grooming, marble burying, and nose poking indicate an increase in RRBs in *Cntnap2*$^{-/-}$ mice. A summary of behavior test results by genotype and sex is shown in **Supplementary file 1**.

To gain further insight into the spontaneous behavior profile of *Cntnap2*$^{-/-}$ mice, we utilized a combination of DeepLabCut and Keypoint-MoSeq to perform unbiased, machine learning-based assessment of general locomotion and behavior in an additional cohort of *Cntnap2*$^{-/-}$ mice (**Figure 5N, P**, **Figure 5—figure supplement 1**; **Mathis et al., 2018**; **Wiltschko et al., 2020**). Again, we found that *Cntnap2*$^{-/-}$ mice did not exhibit changes in general locomotor activity compared to WT littermates (**Figure 5—figure supplement 1A**). Analysis of movement 'syllables' using Keypoint-MoSeq revealed that across the 25 most frequent syllables, two syllables associated with grooming were performed with significantly increased frequency in *Cntnap2*$^{-/-}$ mice (**Figure 5N**). *Cntnap2*$^{-/-}$ mice also had an increase in the total number of grooming bouts (**Figure 5—figure supplement 1B**), replicating the findings in the manually scored cohort (see **Figure 5I**). While syllable usage was generally similar between WT and *Cntnap2*$^{-/-}$ mice, transitions between syllables differed between the groups (**Figure 5—figure supplement 1C**). A measure of the entropy of syllable transitions revealed that *Cntnap2*$^{-/-}$ mice exhibited less entropy, suggesting less variability in the transition from one movement syllable to the next (**Figure 5O**). This rigidity in motor sequence may be indicative of more restricted motor behavior overall. Finally, we tested whether a trained decoder could accurately distinguish WT and *Cntnap2*$^{-/-}$ mice using information about movement, syllable usage, or syllable transitions. The decoding models performed significantly better than chance at identifying WT and *Cntnap2*$^{-/-}$ mice based on their syllable usage and transitions, but not general locomotor activity (**Figure 5P**).

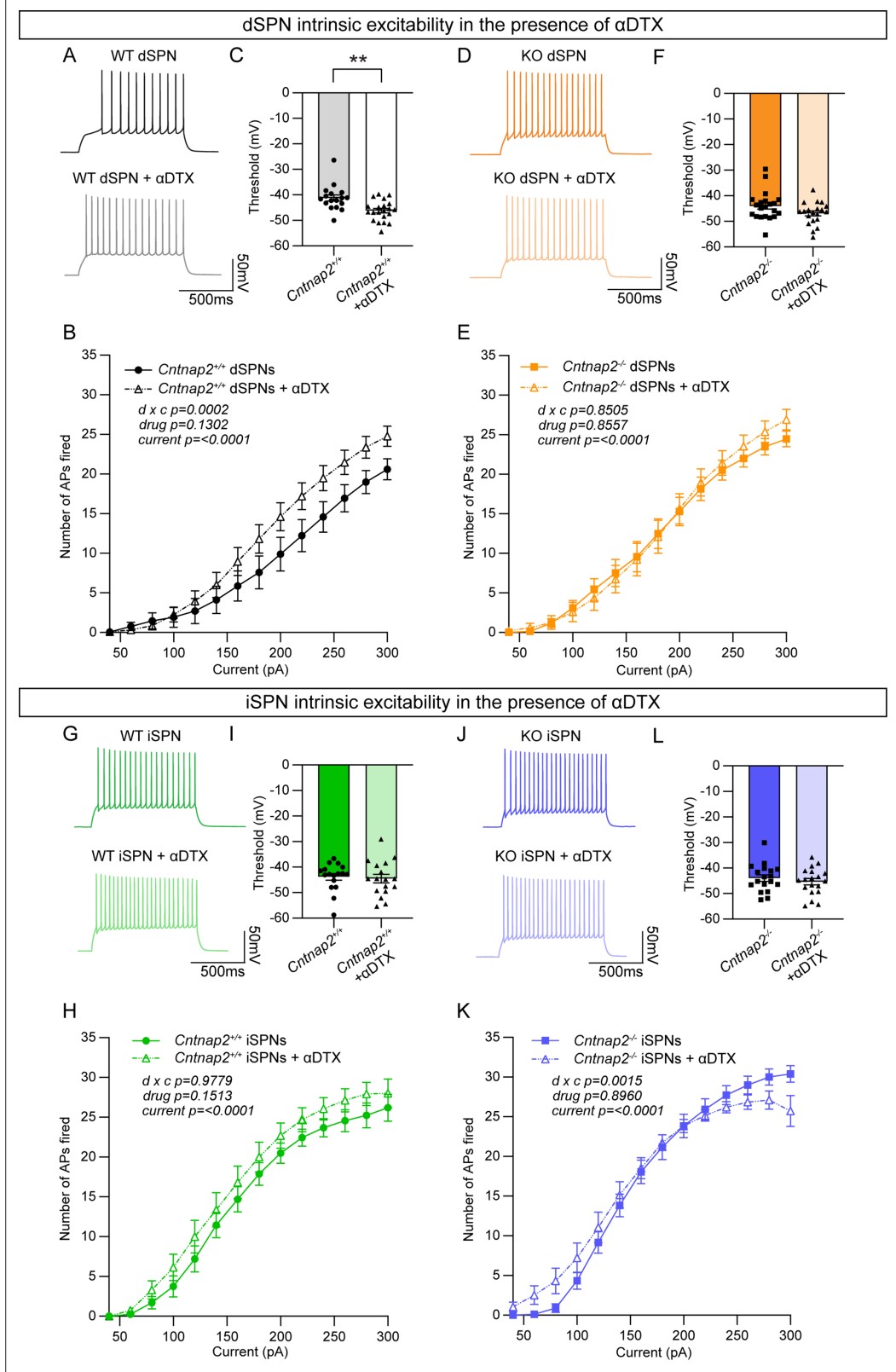

**Figure 4.** The effects of Kv1.2 blockade are occluded in *Cntnap2⁻/⁻* dSPNs. (**A**) Example AP traces from
Cntnap2⁺/⁺ dSPNs evoked by a 200 pA current step in the absence or presence of α-Dendrotoxin (α-DTX).
(**B**) Quantification (mean ± SEM) of the number of APs evoked in *Cntnap2⁺/⁺* dSPNs in the absence or presence of
α-DTX at different current step amplitudes. *Cntnap2⁺/⁺* n = 17 cells from 8 mice, *Cntnap2⁺/⁺* + α-DTX n=21 cells

*Figure 4 continued on next page*

*Figure 4 continued*

from 8 mice. Repeated measures two-way ANOVA p values are shown; drug x c F (13, 468)=3.097, drug F (1, 36)=2.399, current F (2.091, 75.28)=172.0. (**C**) Quantification (mean ± SEM) of the AP threshold in *Cntnap2⁺/⁺* dSPNs in the absence or presence of α-DTX. Dots/triangles represent the threshold for each neuron. n is the same as in panel B. **p=0.0010, Mann-Whitney test. (**D**) Example AP traces from *Cntnap2⁻/⁻* dSPNs evoked by a 200 pA current step in the absence or presence of α-DTX. (**E**) Quantification (mean ± SEM) of the number of APs evoked in *Cntnap2⁻/⁻* dSPNs. *Cntnap2⁻/⁻* n=21 cells from 9 mice, *Cntnap2⁻/⁻* + α-DTX n=20 cells from 9 mice. Repeated measures two-way ANOVA p values are shown; drug x c F (13, 507)=0.6054, drug F (1, 39)=0.03352, current F (2.156, 84.07)=247.8. (**F**) Quantification (mean ± SEM) of the AP threshold in *Cntnap2⁻/⁻* dSPNs. Squares/triangles represent the threshold for each neuron. n is the same as in panel E. p=0.1348, Mann-Whitney test. (**G**) Example AP traces in *Cntnap2⁺/⁺* iSPNs evoked by a 200 pA current step in the absence or presence of α-DTX. (**H**) Quantification (mean ± SEM) of the number of APs evoked in *Cntnap2⁺/⁺* iSPNs. *Cntnap2⁺/⁺* n = 17 cells from 9 mice, *Cntnap2⁺/⁺* + α-DTX n=17 cells from 9 mice. Repeated measures two-way ANOVA p values are shown; drug x c F (13, 416)=0.3719, drug F (1, 32)=2.161, current F (1.955, 62.57)=215.9. (**I**) Quantification (mean ± SEM) of the AP threshold in *Cntnap2⁺/⁺* iSPNs. Dots/triangles represent the threshold for each neuron. n is the same as in panel H. p=0.5401, Mann-Whitney test. (**J**) Example AP traces in *Cntnap2⁻/⁻* iSPNs evoked by a 200 pA current step in the absence or presence of α-DTX. (**K**) Quantification (mean ± SEM) of the number of APs evoked in *Cntnap2⁻/⁻* iSPNs. *Cntnap2⁻/⁻* n=18 cells from 10 mice, *Cntnap2⁻/⁻* + α-DTX n=19 cells from 10 mice. Repeated measures two-way ANOVA p values are shown; drug x c F (13, 455)=2.623, drug F (1, 35)=0.01734, current F (1.721, 60.23)=227.4. (**L**) Quantification (mean ± SEM) of the AP threshold in *Cntnap2⁻/⁻* iSPNs. Squares/triangles represent the threshold for each neuron. n is the same as in panel K. p=0.4250, two-tailed unpaired t test.

The online version of this article includes the following figure supplement(s) for figure 4:

**Figure supplement 1.** Inhibition of Kv1.2 differentially impacts action potential properties in *Cntnap2⁺/⁺* and *Cntnap2⁻/⁻* dSPNs.

**Figure supplement 2.** Inhibition of Kv1.2 does not strongly affect the excitability of iSPNs.

Together, this analysis demonstrates that while overall locomotor activity is not strongly affected in *Cntnap2⁻/⁻* mice, the behavior patterns of these mice are distinct from WT, reflecting enhanced repetitive behaviors.

## Cntnap2⁻/⁻ mice exhibit enhanced motor learning

The accelerating rotarod is a striatal-dependent measure of motor coordination and learning that has been used across a range of ASD mouse models (*Cording and Bateup, 2023*). Changes in corticostriatal circuits have been identified in mouse models of ASD with altered performance in the task (*Cording and Bateup, 2023*). Given the altered corticostriatal drive in *Cntnap2⁻/⁻* mice, we tested whether motor coordination and learning were affected in these mice. In the rotarod test, mice learn to walk and then run to stay on a rotating rod as it increases in speed over the course of 5 min. Mice perform three trials a day for 4 days. In trials one through six, the rod increases in speed from 5 to 40 revolutions per minute (RPM), while in trials 7 through 12, the rod increases from 10 to 80 RPM (*Figure 6A*). Learning occurs over trials within a day, as well as across days, as the mouse develops and hones a stereotyped motor pattern to stay on the rod for increasing amounts of time (*Rothwell et al., 2014*; *Yin et al., 2009*). We found that *Cntnap2⁻/⁻* mice performed significantly better than WT mice in this task, particularly in the later trials when the rod rotates at the faster 10–80 RPM speed (*Figure 6B*). Initial performance (terminal velocity on trial one) was not significantly different between WT and *Cntnap2⁻/⁻* mice (*Figure 6C*), but the learning rate from trial one to trial 12 was increased in *Cntnap2⁻/⁻* mice (*Figure 6D*). These findings expand upon previous work indicating increased performance on both steady-state and accelerating rotarod tasks utilizing slower speeds in *Cntnap2⁻/⁻* mice (*Dawes et al., 2018*; *Peñagarikano et al., 2011*). These results also align with the increased rotarod performance seen in other ASD mouse models exhibiting enhanced corticostriatal drive (*Benthall et al., 2021*; *Cording and Bateup, 2023*).

## Cntnap2⁻/⁻ mice exhibit cognitive inflexibility

RRBs include not just stereotyped movements, but also insistence on sameness and perseverative interests (*APA, 2022*). Cognitive inflexibility, a deficit in the ability to flexibly adapt to changes in the environment and update behavior, is a manifestation of ASD and some other psychiatric disorders, which are associated with striatal dysfunction (*Fuccillo, 2016*). Indeed, in individuals with ASD, the

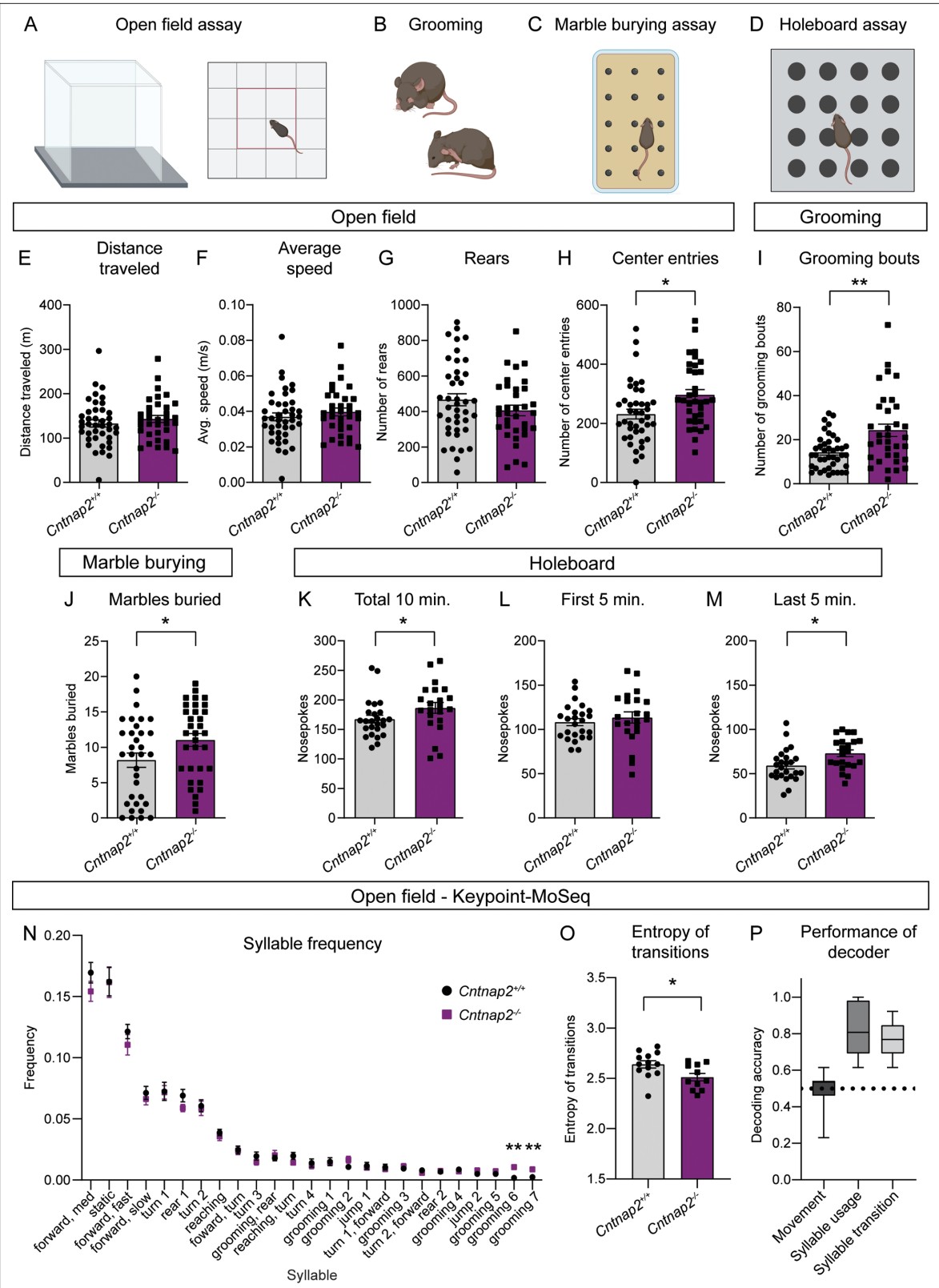

**Figure 5.** *Cntnap2^{-/-}* mice have increased repetitive behaviors. (**A–D**) Schematics of the behavioral assays used to measure repetitive behaviors in *Cntnap2^{+/+}* and *Cntnap2^{-/-}* mice. Created with BioRender.com. Male and female mice were used for all tests. (**E–H**) Quantification (mean ± SEM) of open field activity over 60 min. (**E**) Total distance traveled, p=0.3538, Mann-Whitney test; (**F**) average speed, p=0.3832, Mann-Whitney test; (**G**) number of rears, p=0.1892, two-tailed unpaired t test; (**H**) number of center entries, *p=0.0101, two-tailed unpaired t test. *Cntnap2^{+/+}* n = 41 mice, *Cntnap2^{-/-}*

*Figure 5 continued on next page*

*Figure 5 continued*

n=34 mice. (**I**) Quantification (mean ± SEM) of the number of manually scored grooming bouts in the first 20 min of the open field test, \*\*p=0.0034, Mann-Whitney test, *Cntnap2+/+* n = 41 mice, *Cntnap2-/-* n=34 mice. (**J**) Quantification (mean ± SEM) of total marbles buried in the marble burying assay. *Cntnap2+/+* n = 33 mice and *Cntnap2-/-* n=33 mice, \*p=0.0396, two-tailed unpaired t test. (**K–M**) Quantification (mean ± SEM) of performance in the holeboard assay. (**K**) Total number of nose pokes in 10 min, \*p=0.0212, Mann-Whitney test; (**L**) nose pokes in the first 5 min, p=0.4811, two-tailed unpaired t test; and (**M**) nose pokes in the last 5 min, \*p=0.0116, two-tailed unpaired t test. *Cntnap2+/+* n = 25 mice, *Cntnap2-/-* n=22 mice. (**N**) Quantification (mean ± SEM) of the frequency of movement syllables (top 25 most frequent syllables) in the open field assay defined by Keypoint-MoSeq. *Cntnap2+/+* n = 13 mice and *Cntnap2-/-* n=11 mice, \*\*p=0.0013 for grooming 6, \*\*p=0.0013 for grooming 7, Kruskal-Wallis test with Dunn's correction for multiple comparisons. (**O**) Quantification (mean ± SEM) of the entropy of syllable transitions in the open field assay. *Cntnap2+/+* n = 13 mice and *Cntnap2-/-* n=11 mice, \*p=0.0236, two-tailed unpaired t test. (**P**) Accuracy of a Random Forest decoder trained on DeepLabCut basic locomotor data (Movement), Keypoint-MoSeq syllable usage data (Syllable usage), or Keypoint-MoSeq syllable transition data (Syllable transition) in distinguishing between *Cntnap2+/+* and *Cntnap2-/-* mice. Dotted line represents chance performance. For panels E-M and O, dots/squares represent the value for each mouse.

The online version of this article includes the following figure supplement(s) for figure 5:

**Figure supplement 1.** DeepLabCut and Keypoint-MoSeq analysis of *Cntnap2-/-* mice.

severity of RRBs is associated with measures of cognitive inflexibility, and evidence from imaging studies suggests that altered corticostriatal connectivity may be present in the case of both repetitive behaviors and cognitive inflexibility (*Uddin, 2021*). To assess cognitive flexibility in *Cntnap2-/-* mice, we utilized a four-choice odor-based reversal learning assay (*Johnson et al., 2016*; *Lin et al., 2022*). Briefly, mice were trained to dig for a food reward in one of four pots containing scented wood shavings (*Figure 7A*). On the first day of the task (acquisition), the rewarded pot was scented with odor one (O1). Mice reached the criterion when they chose O1 for at least eight of 10 consecutive trials. On day two, mice were given a recall test in which the rewarded pot was again scented with O1. After reaching the criterion, the reversal trials began, and the rewarded pot was scented with the previously unrewarded odor two (O2). To reach the criterion, mice must learn the new association of O2 and reward and choose O2 for eight of 10 consecutive trials.

During acquisition, *Cntnap2-/-* mice performed similarly to WT controls, not differing in the average number of trials needed to reach criterion, the number of quadrant entries made before making a choice, or the latency to choose a pot (*Figure 7B–D*). On day 2, *Cntnap2-/-* mice exhibited typical recall, demonstrating successful consolidation of the odor-reward pairing (*Figure 7E*). However, we

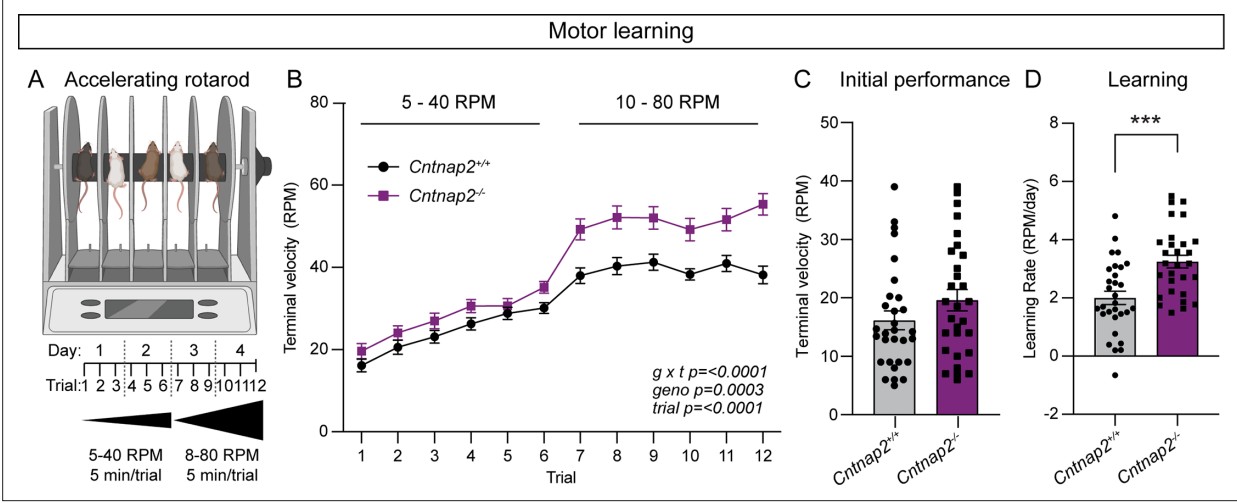

**Figure 6.** *Cntnap2-/-* mice exhibit enhanced motor learning. (**A**) Schematic of the rotarod apparatus (top), and design of the task (bottom). Created with BioRender.com. Mice walk to stay on the rotating rod for three 5-min trials a day for 2 days at 5–40 RPM acceleration over 5 min, followed by three trials a day for 2 days at 10–80 RPM. (**B**) Quantification (mean ± SEM) of accelerating rotarod performance across 12 trials for the indicated genotypes. *Cntnap2+/+* n = 30 mice, *Cntnap2-/-* n=29 mice. Repeated measures two-way ANOVA p values are shown; g x t F (11, 616)=4.935, geno F (1, 56)=15.29, trial F (7.245, 405.7)=108.4. (**C**) Quantification (mean ± SEM) of rotarod performance on trial 1 quantified as terminal speed. Dots/squares represent the performance of individual mice. n is same as in panel B, p=0.1518, Mann-Whitney test. (**D**) Quantification (mean ± SEM) of learning rate (RPM/day) calculated as the slope of the line of performance from the first trial (1) to the last trial (12) for each mouse. Dots/squares represent the learning rate for individual mice. n is the same as in panel B, \*\*\*p=0.0002, two-tailed unpaired t test.

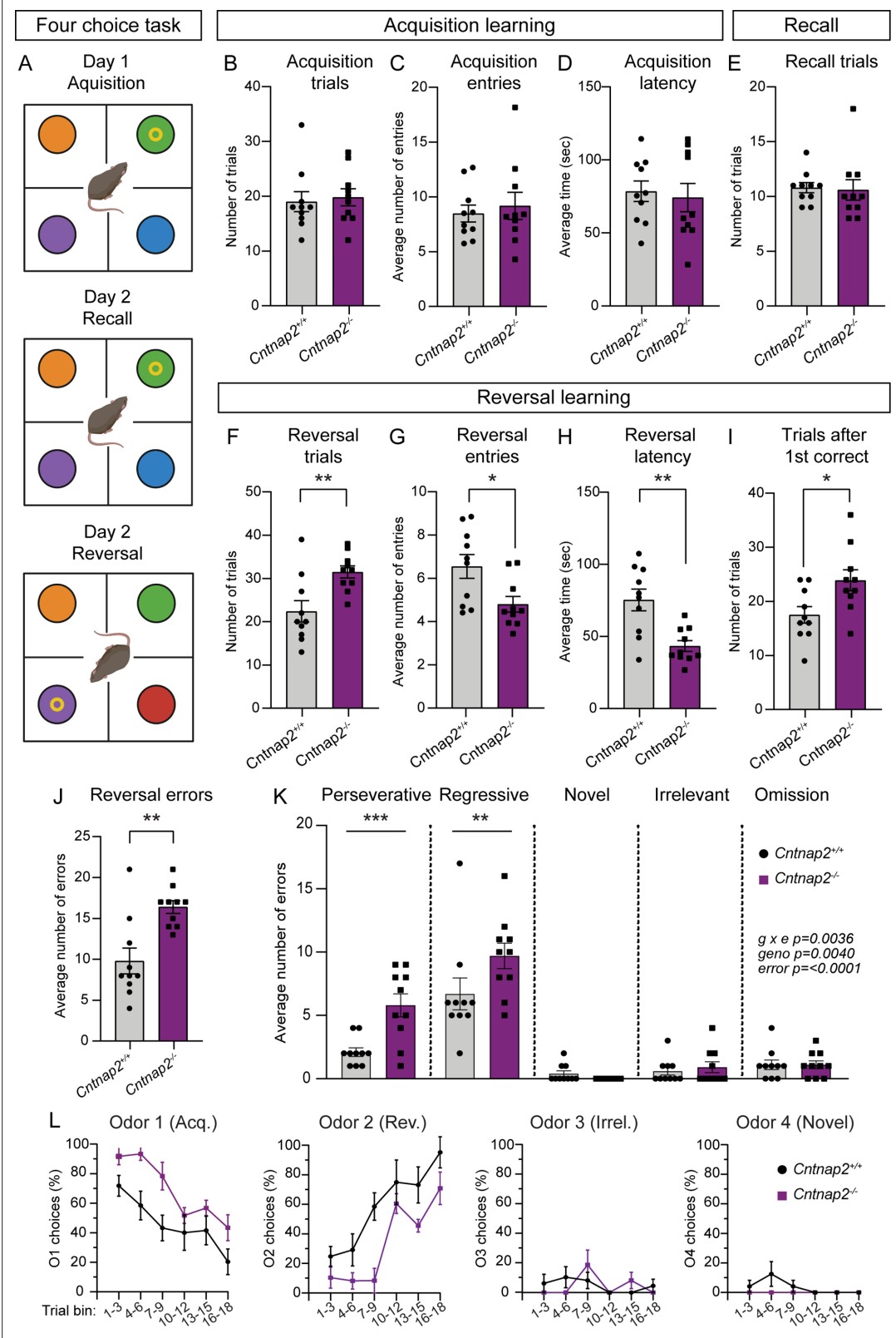

**Figure 7.** *Cntnap2−/−* mice demonstrate cognitive inflexibility. (**A**) Schematic of the four-choice odor-based reversal learning task. Created with BioRender.com. Colored circles represent pots with different scented wood shavings. Yellow ring represents the food reward. Red circle in the Day 2 Reversal panel indicates a novel odor. (**B–D**) Quantification of parameters during acquisition learning. Mean ± SEM number of trials to reach criterion (B, at least 8 out of last 10 trials correct), p=0.5397, Mann-Whitney test; number of quadrant entries before making a choice (**C**), p=0.9118, Mann-Whitney

*Figure 7 continued on next page*

*Figure 7 continued*

test; and latency to make a choice (**D**), p=0.7224, two-tailed unpaired t test. Dots/squares represent the value for each mouse. (**E**) Quantification (mean ± SEM) of the number of trials to reach criterion (at least 8 out of last 10 trials correct) during the recall test on day 2, p=0.3737, Mann-Whitney test. (**F–I**) Quantification of parameters during reversal learning. Mean ± SEM number of trials to reach criterion (F, at least 8 out of last 10 trials correct), \*\*p=0.0048, two-tailed unpaired t test; number of quadrant entries before making a choice (**G**), \*p=0.0158, two-tailed unpaired t test; latency to make a choice (**H**), \*\*p=0.0013, two-tailed unpaired t test; and number of trials to reach criterion after the first correct choice (**I**), \*p=0.0183, two-tailed unpaired t test. (**J**) Quantification (mean ± SEM) of the total number of errors made during reversal learning, \*\*p=0.0034, Mann-Whitney test. (**K**) Quantification (mean ± SEM) of the different error types made during reversal learning. Perseverative errors, \*\*\*p=0.0005, regressive errors \*\*p=0.0068, novel errors, p=0.9955, irrelevant errors p=0.9988, omissions, p=>0.9999, repeated measures two-way ANOVA with Šídák's multiple comparisons test; g x e F (4, 72)=4.292, geno F (1, 18)=10.91, error F (4, 72)=53.49. (**L**) Quantification (mean ± SEM) of the percent of choices made for each odor, binned across three trials, during reversal learning. Odor 1 was rewarded during acquisition learning. Odor 2 was rewarded during reversal learning. Odor 3 was never rewarded (irrelevant). Odor 4 was a novel odor introduced during the reversal learning phase. For panels B-L, n=10 *Cntnap2*$^{+/+}$ mice and 10 *Cntnap2*$^{-/-}$ mice.

found that *Cntnap2*$^{-/-}$ mice had a deficit in reversal learning, requiring significantly more trials on average than WTs to reach criterion once the odor-reward pairing was changed (*Figure 7F*). Interestingly, during reversal, *Cntnap2*$^{-/-}$ mice made fewer quadrant entries before making a digging choice and had significantly decreased latency to make a choice compared to controls (*Figure 7G and H*). Even after the first correct choice of O2 during reversal, *Cntnap2*$^{-/-}$ mice took more trials to reach criterion than WTs (*Figure 7I*). In terms of errors, *Cntnap2*$^{-/-}$ mice made more reversal errors than WT mice (*Figure 7J*), in particular perseverative (continuing to choose O1) and regressive (choosing O1 after correctly choosing O2 once) errors (*Figure 7K*). *Cntnap2*$^{-/-}$ mice did not differ from WT controls in choices of the novel (newly introduced during reversal) or irrelevant (never rewarded) odors, or in the number of omitted trials (timing out without making a choice; *Figure 7K*). Instead, the persistence in choosing O1, even after at least one correct choice of O2, drove the cognitive inflexibility in these mice (*Figure 7L*). This persistence in choice may be reflective of the broader scope of RRBs in *Cntnap2*$^{-/-}$ mice.

## Discussion

In this study, we tested whether loss of the neurodevelopmental disorder risk gene *Cntnap2* altered striatal physiology or striatal-dependent behaviors. We found that direct pathway SPNs exhibited enhanced cortical drive in *Cntnap2*$^{-/-}$ mice. This change was not due to differences in excitatory or inhibitory synapses, as cortical inputs onto SPNs were unchanged and there were no significant deficits in inhibition onto SPNs in these mice. Instead, loss of *Cntnap2* resulted in a significant increase in the excitability of dSPNs, likely driven by altered contribution of Kv1.2 channels to intrinsic firing properties. At the behavioral level, *Cntnap2*$^{-/-}$ mice exhibited repetitive behaviors including increased grooming, nose poking, and marble burying. These mice also had enhanced motor learning, performing significantly better than controls in the accelerating rotarod task. Finally, *Cntnap2*$^{-/-}$ mice exhibited cognitive inflexibility in the four-choice reversal learning assay.

### Cellular phenotypes of *Cntnap2* loss

The loss of Caspr2 has a variable impact on intrinsic excitability across brain regions and cell types. The increased intrinsic excitability that we identified in dSPNs has also been observed in Purkinje cells of the cerebellum (*Fernández et al., 2021*) and pyramidal cells of the cortex (*Antoine et al., 2019*; *Cifuentes-Diaz et al., 2023*). However, we note hypoactivity (*Brumback et al., 2018*) or unchanged excitability (*Lazaro et al., 2019*) of pyramidal cells in some cortical regions in *Cntnap2*$^{-/-}$ mice. Caspr2 is involved in the clustering of voltage-gated potassium channels, in particular at the juxtaparanodes of myelinated axons (*Poliak et al., 1999*; *Poliak et al., 2003*) and the axon initial segment (*Inda et al., 2006*). Indeed, there are profound deficits in the clustering of Kv1-family channels in *Cntnap2*$^{-/-}$ mice, particularly Kv1.2 channels (*Scott et al., 2019*). These channels play an important role in regulating the intrinsic excitability of SPNs (*Nisenbaum et al., 1994*), in particular dSPNs (*Lahiri and Bevan, 2020*), and when blocked, result in increased excitability (*Shen et al., 2004*). The loss of Caspr2 in *Cntnap2*$^{-/-}$ mice may result in improper localization of Kv1.2 channels, and thus alter their contribution to intrinsic excitability. Indeed, we found that the effects of Kv1.2 blockade were largely occluded in *Cntnap2*$^{-/-}$ dSPNs, indicating that Kv1.2 loss of function is likely the mechanism driving the change in

excitability. Interestingly, we find that blocking Kv1.2 channels has less of an effect on the excitability of iSPNs, which may account for the greater impact of *Cntnap2* loss on dSPN physiology.

Prior studies of *Cntnap2⁻/⁻* mice have identified changes in the number of PV-expressing interneurons in the cortex (*Peñagarikano et al., 2011*; *Vogt et al., 2018*), hippocampus (*Paterno et al., 2021*; *Peñagarikano et al., 2011*), and striatum (*Peñagarikano et al., 2011*). However, this finding is inconsistent across studies, as others have reported no change in the number of PV interneurons in these regions (*Ahmed et al., 2023*; *Lauber et al., 2018*; *Scott et al., 2019*). One possible explanation for this disparity is altered PV protein expression in *Cntnap2⁻/⁻* mice such that immunoreactivity varies in cell counting assessments. This is supported by the finding that the number of Vicia Villosa Agglutinin-positive (VVA+) perineuronal nets that preferentially surround PV cells is unchanged in *Cntnap2⁻/⁻* mice, even when PV immunoreactivity varies (*Härtig et al., 1992*; *Haunsø et al., 2000*; *Lauber et al., 2018*). Parvalbumin, a $Ca^{2+}$ buffer, plays an important role in the intrinsic fast-spiking properties of PV interneurons, such that a reduction in PV protein expression is known to change PV intrinsic function (*Orduz et al., 2013*). However, altered intrinsic properties of PV interneurons have also been variably reported across brain regions and studies of *Cntnap2⁻/⁻* mice, with subtle changes in PV firing properties reported in the developing striatum (*Ahmed et al., 2023*) and adult cortex (*Vogt et al., 2018*), but unchanged in the hippocampus (*Paterno et al., 2021*) and medial prefrontal cortex (*Lazaro et al., 2019*). In this study, we find no significant change in the number of PV interneurons or the striatal expression of PV protein in *Cntnap2⁻/⁻* mice. Consistent with this, we find no deficits in PV-mediated inhibition onto SPNs. Together, this suggests that primary changes in PV interneurons are unlikely to account for altered striatal circuit function in *Cntnap2⁻/⁻* mice.

## Loss of *Cntnap2* alters striatal-dependent behaviors

The striatum can be separated into functionally distinct subregions. We focused on the dorsal striatum in this study because of its role in controlling motor and cognitive functions (*Voorn et al., 2004*), which are relevant to ASD (*Fuccillo, 2016*; *Subramanian et al., 2017*). The dorsal striatum can be further subdivided into the dorsomedial striatum (DMS) and the DLS, with the former considered an associative region involved in goal-directed action-outcome learning and the latter implicated in the acquisition of habitual or procedural behaviors (*Packard and Knowlton, 2002*). We focused on cellular properties in the DLS as stereotyped, perseverative, or persistent behaviors likely recruit DLS circuitry (*Evans et al., 2024*; *Fuccillo, 2016*). In the accelerating rotarod assay, learning and performance in the task have been associated with changes in the DLS. Positive modulation of the firing rate of DLS neurons occurs during rotarod training, in particular in later trials of the task, and synaptic potentiation of DLS SPNs in late training is necessary for intact performance (*Yin et al., 2009*). In line with this, lesions of the DLS impair both early and late rotarod learning (*Yin et al., 2009*). We found that *Cntnap2⁻/⁻* mice had increased rotarod performance, most notably at the later stages when DLS function is strongly implicated. Functionally, we also found increased cortical drive of DLS dSPNs in these mice, a change that was sufficient to increase rotarod performance in another mouse model with disruption of an ASD-risk gene (*Benthall et al., 2021*). Together, this supports a connection between the change observed in DLS SPN physiology and the increased motor routine learning in *Cntnap2⁻/⁻* mice, although this idea remains to be causally tested.

In terms of restricted, repetitive behaviors, we replicated prior studies showing increased spontaneous grooming in *Cntnap2⁻/⁻* mice (*Peñagarikano et al., 2011*). Early evidence implicates the striatum in the control of the syntax or sequence of movements in a rodent grooming bout, such that very small lesions of DLS are capable of disrupting grooming (*Cromwell and Berridge, 1996*). However, recent work has also outlined roles for cellular modulation in DMS and ventral striatal Islands of Calleja in the control of grooming behavior (*Ramírez-Armenta et al., 2022*; *Zhang et al., 2021*). *Cntnap2⁻/⁻* mice also exhibited increased marble burying and nose poking. The precise neurobiological substrates of these behaviors are yet unclear, but evidence linking increases in these behaviors to changes in cortico-striatal and amygdala-striatal function supports the notion that these behaviors may fit into a broader basal ganglia-associated RRB-like domain (*Albelda and Joel, 2012*; *Lee et al., 2024*).

In the four-choice reversal learning task, *Cntnap2⁻/⁻* mice showed no differences during the acquisition phase, suggesting that there were no broad deficits in reward learning. However, in the reversal stage of the task, *Cntnap2⁻/⁻* mice took significantly more trials to learn a new odor-reward pairing, owing primarily to continued choice of the previously rewarded odor. The DMS and ventral striatum

(nucleus accumbens) have been shown to play an important role in reversal learning (*Izquierdo et al., 2017*), and in the four-choice task specifically (*Delevich et al., 2022*). Additionally, decreased dopamine release in the DLS is associated with deficits in reversal learning in this task (*Kosillo et al., 2019*; *Lin et al., 2022*). Together, the learning phenotypes seen in *Cntnap2⁻/⁻* mice in the accelerating rotarod and reversal learning assay share an underlying rigidity in behavioral choice. In both cases, changes in striatal circuits likely underlie the repetitive, stereotyped behaviors.

In summary, our results fit into a model whereby divergent cellular changes in the striatum driven by a functionally diverse set of ASD risk genes similarly enhance corticostriatal drive, in particular, of the direct pathway. This, in turn, may facilitate striatal-dependent motor routine learning and behavioral perseveration. We speculate that a shared gain-of-function in striatal circuits may play a role in the formation of perseverative or repetitive behaviors in a sub-set of ASDs more broadly.

## Limitations and future directions

This study characterized multiple striatal cell types and synapses as well as striatum-associated behaviors in *Cntnap2⁻/⁻* mice for the first time. However, there remain open questions as to the impact that *Cntnap2* loss has on striatal function. Although we did not identify excitatory or inhibitory synaptic changes in *Cntnap2⁻/⁻* SPNs in this study, we only focused on a subset of these connections. While cortical inputs are a major source of excitation onto SPNs, there are other excitatory inputs onto these cells, such as from the thalamus, that were not assessed in this study (*Ding et al., 2008*; *Doig et al., 2010*; *Gerfen and Surmeier, 2011*). In addition, while all intrastriatal GABAergic interneurons would have been sampled in the electrical stimulation experiments in this study, it is possible that interrogation of individual interneuron subtypes would reveal changes in specific inhibitory connections. Finally, major modulators of SPN activity such as cholinergic interneurons and dopaminergic inputs were not assessed here. These inputs have been implicated in several ASD mouse models (*Kosillo and Bateup, 2021*; *Pavăl and Micluția, 2021*; *Rapanelli et al., 2017*), and changes in striatal cholinergic interneuron function have been identified in young (P21) *Cntnap2⁻/⁻* mice, making further study of this circuit particularly cogent (*Ahmed et al., 2023*).

The major cellular phenotype we observed was enhanced intrinsic excitability of *Cntnap2⁻/⁻* dSPNs. Given that Kv1.2 channels are known to be organized in part by Caspr2 (*Poliak et al., 1999*; *Poliak et al., 2003*; *Scott et al., 2019*) and the fact that blockade of Kv1.2 did not affect the excitability of *Cntnap2⁻/⁻* dSPNs, we conclude that loss of *Cntnap2* leads to the improper clustering, number, or function of Kv1.2. However, more direct measurement of the number and/or localization of these channels through imaging, or the function of these channels through voltage clamp measurement of potassium currents, would bolster this conclusion. Further, assessing whether similar occlusion of the effects of blocking Kv1.2 channels occurs in the cortical and cerebellar cell types that have also been shown to be hyperexcitable in *Cntnap2⁻/⁻* mice would support a more holistic understanding of the impact of *Cntnap2* loss on neuronal function (*Antoine et al., 2019*; *Cifuentes-Diaz et al., 2023*; *Fernández et al., 2021*). Considering that these cell types also exhibit α-DTX-sensitive currents that boost AP firing, it is possible that the mechanism by which *Cntnap2* loss increases excitability may be shared (*Guan et al., 2007*; *Haghdoust et al., 2007*; *Khavandgar et al., 2005*).

Finally, while we did identify changes in several striatum-associated behaviors in *Cntnap2⁻/⁻* mice, a causative relationship between the physiological and behavioral changes that we identified has not been established. Given that hyperexcitability of the SPNs of the movement-initiating direct pathway is the primary physiological change we identified, testing whether this change is necessary (i.e. by decreasing dSPN activity in *Cntnap2⁻/⁻* mice through cell-type-specific expression of inward rectifying potassium channel Kir2.1) or sufficient (i.e. by increasing dSPN activity in WT mice using cell-type-specific $G_q$-coupled (hM2Dq) DREADD activation) to alter behavior could illuminate the relationship between striatal function and behavior in *Cntnap2⁻/⁻* mice.

## Materials and methods
### Mice

All animal procedures were conducted in accordance with protocols approved by the University of California, Berkeley Institutional Animal Care and Use Committee (IACUC) and Office of Laboratory

Animal Care (OLAC) (AUP-2016-04-8684-3). *Table 1* lists the mouse lines used for each experiment and their source.

Mice were group housed on a 12 hr light/dark cycle (dark cycle 9:00 AM – 9:00 PM) and given ad libitum access to standard rodent chow and water. Both male and female animals were used for experimentation. The ages, sexes, and numbers of mice used for each experiment are indicated in the respective method details and figure legends. All mice used for experiments were heterozygous or hemizygous for the *Drd1a*-tdTomato, *Thy1*-ChR2-YFP, PV-Cre, Ai32, or Ai9 transgenes to avoid potential physiological or behavioral alterations.

## Electrophysiology

Mice (P50-60) were briefly anesthetized with isoflurane and perfused transcardially with ice-cold ACSF (pH = 7.4) containing (in mM): 127 NaCl, 25 NaHCO3, 1.25 NaH2PO4, 2.5 KCl, 1 MgCl2, 2 CaCl2, and 25 glucose, bubbled continuously with carbogen (95% $O_2$ and 5% $CO_2$). Brains were rapidly removed, and coronal slices (275 µm) were cut on a VT1000S vibratome (Leica) in oxygenated ice-cold choline-based external solution (pH = 7.8) containing (in mM): 110 choline chloride, 25 NaHCO3, 1.25 NaHPO4, 2.5 KCl, 7 MgCl2, 0.5 CaCl2, 25 glucose, 11.6 sodium ascorbate, and 3.1 sodium pyruvate. Slices were recovered in ACSF at 36 °C for 15 min and then kept at room temperature (RT) before recording. Recordings were made with a MultiClamp 700B amplifier (Molecular Devices) at RT using 3–5 MOhm glass patch electrodes (Sutter, #BF150-86-7.5). Data were acquired using ScanImage software, written and maintained by Dr. Bernardo Sabatini (https://github.com/bernardosabatini/SabalabAcq, *Sabatini, 2022*). Traces were analyzed in Igor Pro (Wavemetrics). Recordings with a

**Table 1.** Summary of mouse lines used.

| Experiment | Mouse line | Allele 1 (reference; JAX strain #) | Allele 2 (reference; JAX strain #) | Allele 3 (reference; JAX strain #) | Allele 4 (reference; JAX strain #) |
|---|---|---|---|---|---|
| Corticostriatal transmission (*Figure 1*) | *Cntnap2*;D1-tdTomato;Thy1-ChR2 | *Cntnap2* (*Poliak et al., 2003*; #017482) | *Drd1a*-tdTomato (*Ade et al., 2011*; #016204) | *Thy1*-ChR2-YFP (*Arenkiel et al., 2007*; #007612) | |
| General inhibition (*Figure 2A–E*) | *Cntnap2*;D1-tdTomato | *Cntnap2* (*Poliak et al., 2003*; #017482) | *Drd1a*-tdTomato (*Ade et al., 2011*; #016204) | | |
| PV-specific inhibition (*Figure 2F–J*) | *Cntnap2*;D1-tdTomato;PV-Cre;Ai32 | *Cntnap2* (*Poliak et al., 2003*; #017482) | *Drd1a*-tdTomato (*Ade et al., 2011*; #016204) | *Pvalb*-Cre (*Hippenmeyer et al., 2005*; #017320) | Ai32 (*Madisen et al., 2012*; #012569) |
| SPN intrinsic excitability (*Figure 3*, *Figure 3—figure supplement 1*) | *Cntnap2*;D1-tdTomato | *Cntnap2* (*Poliak et al., 2003*; #017482) | *Drd1a*-tdTomato (*Ade et al., 2011*; #016204) | | |
| SPN intrinsic excitability in the presence of α-DTX (*Figure 4*, *Figure 4—figure supplement 1*, *Figure 4—figure supplement 2*) | *Cntnap2*;D1-tdTomato | *Cntnap2* (*Poliak et al., 2003*; #017482) | *Drd1a*-tdTomato (*Ade et al., 2011*; #016204) | | |
| Behavior experiments* (*Figures 5–7*, *Figure 5—figure supplement 1*) | *Cntnap2*;D1-tdTomato | *Cntnap2* (*Poliak et al., 2003*; #017482) | *Drd1a*-tdTomato (*Ade et al., 2011*; #016204) | | |
| Spine analysis, PV cell counting (*Figure 1—figure supplement 1*, *Figure 2—figure supplement 2A–H*) | *Cntnap2*;D1-tdTomato | *Cntnap2* (*Poliak et al., 2003*; #017482) | *Drd1a*-tdTomato (*Ade et al., 2011*; #016204) | | |
| PV intrinsic excitability (*Figure 2—figure supplement 1*) | *Cntnap2*;PV-Cre;Ai9 | *Cntnap2* (*Poliak et al., 2003*; #017482) | *Pvalb*-Cre (*Hippenmeyer et al., 2005*; #017320) | Ai9 (*Madisen et al., 2010*; #007909) | |
| PV in situ, western blot (*Figure 2—figure supplement 2I–N*) | *Cntnap2* | *Cntnap2* (*Poliak et al., 2003*; #017482) | | | |

*Littermate animals both positive and negative for D1-tdTomato were used in behavior experiments.

series resistance >25 MOhms or holding current more negative than –200 pA were rejected. Passive properties were calculated using the double exponential curve fit of the average of five –5 mV, 100ms long pulse steps applied at the beginning of every experiment.

## Current-clamp recordings

Current clamp recordings were made using a potassium-based internal solution (pH = 7.4) containing (in mM): 135 KMeSO4, 5 KCl, 5 HEPES, 4 Mg-ATP, 0.3 Na-GTP, 10 phosphocreatine, and 1 EGTA. For corticostriatal excitability experiments, optogenetic stimulation consisted of a full-field pulse of blue light (470 nm, 0.5ms pulse width, CoolLED) through a 63 x objective (Olympus, LUMPLFLN60XW). Light power was linear over the range of intensities tested. No synaptic blockers were included. For intrinsic excitability experiments (SPN, PV interneuron, and SPN + α-DTX experiments), NBQX (10 μM, Tocris, #1044), CPP (10 μM, Tocris, #0247), and picrotoxin (50 μM, Abcam, #120315) were added to the external solution to block synaptic transmission. For Kv1.2 inhibition experiments, α-DTX (100 nM, Alomone Labs, #D-350) was added to the external solution. Control recordings in the absence of α-DTX were performed on slices prior to drug application or on fresh slices after drug washout in alternating order across recording days. Bovine serum albumin (BSA, 0.005%, Sigma, #A7030) was included in both control and α-DTX-containing external solutions to minimize nonspecific binding. One-second depolarizing current steps were applied to induce APs. No holding current was applied to the membrane.

## Voltage-clamp recordings

Voltage-clamp recordings were made using a cesium-based internal solution (pH = 7.4) containing (in mM): 120 CsMeSO4, 15 CsCl, 10 TEA-Cl, 8 NaCl, 10 HEPES, 1 EGTA, 5 QX-314, 4 Mg-ATP, and 0.3 Na-GTP. Recordings were acquired with the amplifier Bessel filter set at 3 kHz. Corticostriatal synaptic stimulation experiments to measure evoked EPSCs were performed in picrotoxin (50 μM), and optogenetic stimulation consisted of a full-field pulse of blue light (470 nm, 0.15ms pulse width) through a 63 x objective. To record AMPAR-mediated EPSCs, cells were held at –70 mV; to record NMDAR-mediated EPSCs, cells were held at +40 mV. Synaptic stimulation experiments to measure evoked IPSCs were performed in NBQX (10 μM) and CPP (10 μM). For electrically evoked IPSCs, a concentric bipolar stimulating electrode (FHC, #30214) was placed in dorsal striatum, roughly 200 μm medial to the recording site in DLS, and a 0.15ms stimulus was applied. For PV-interneuron optically evoked IPSCs, a full-field pulse of blue light (470 nm, 0.15ms pulse width) was applied through a 63 x objective at the recording site. All evoked IPSC experiments were recorded with cells held at –70 mV.

## Dendritic imaging and spine analysis

Neonatal (P1-3) *Cntnap2*$^{-/-}$;D1-tdT and *Cntnap2*$^{+/+}$;D1-tdT mice were cryoanesthetized and injected bilaterally with 200 nL AAV1.hSyn.eGFP.WPRE.bGH (Penn Vector Core, #p1696 *Keaveney et al., 2018*), diluted 1:75 in saline to achieve sparse transduction. Injections were targeted to the dorsal striatum, with coordinates approximately 1.3 mm lateral to midline, 2.0 mm posterior to bregma, and 1.5 mm ventral to the head surface. At P50-60, mice were anesthetized with isoflurane and transcardial perfusion was performed with 10 mL of 1 x PBS followed by 10 mL of ice-cold 4% PFA (EMS, #15,710 S) in 1 x PBS. Brains were post-fixed in 4% PFA in 1 x PBS overnight at 4 ° C. 80 μm coronal sections were made using a freezing microtome (American Optical, AO 860) and stored in 1 x PBS at 4 ° C. Sections were blocked for 1 hr at RT in BlockAid (ThermoFisher, #B10710) and incubated for 48 hr with gentle shaking at 4 ° C with antibodies against GFP (1:2500, Abcam, #13970) and RFP (1:1000, Rockland VWR, #600-401-379) diluted in PBS-Tx 1 x PBS with 0.25% Triton X-100 (Sigma, #T8787). Sections were washed 3x10 min in PBS-Tx and incubated with gentle shaking for 1 hr at RT with Alexa Fluor 488 and 546 secondary antibodies (1:500, Invitrogen, #A11039, #A11035). Sections were washed 3x10 min in 1 x PBS and mounted onto SuperFrost slides (VWR, #48311–703) using VECTASHIELD HardSet Antifade Mounting Medium (Vector Laboratories, #H-1400–10). Z-stack images of individual dendrites were taken on a confocal microscope (Olympus FLUOVIEW FV3000) with a 60 x oil immersion objective (Olympus #1-U2B832) at 2.5 x zoom with a step size of 0.4 μm and deconvoluted using Olympus CellSens software. To quantify spine density, dendrites and spines were reconstructed using the FilamentTracer module in Imaris software (Oxford Instruments). The spine

density of each dendrite was calculated using Imaris. Dendrites analyzed varied in total length, but excluded the most proximal and distal portions of the dendrite.

## Brain sectioning and immunohistochemistry

Adult mice were perfused as above, and brains were post-fixed with 4% paraformaldehyde overnight, then sectioned coronally at 30 μm. For immunohistochemistry, individual wells of sections were washed for 3x5 min with 1 x PBS, then blocked for 1 hr at RT with BlockAid blocking solution. Primary antibodies diluted in PBS-Tx were added, and tissue was incubated for 48 hr with gentle shaking at 4 ° C. Sections were then washed 3x10 min with PBS-Tx. Secondary antibodies diluted 1:500 in PBS-Tx were added and incubated with gentle shaking for 1 hr at RT. Sections were washed 3x10 min in 1 x PBS. Sections were mounted onto SuperFrost slides (VWR, #48311–703) and coverslipped with VECTASHIELD HardSet with DAPI (Vector Laboratories, #H-1500–10) or VECTASHIELD HardSet Antifade Mounting Medium (Vector Laboratories, #H-1400–10). The following antibodies were used: mouse anti-PV (1:1000, Sigma, #P3088), rabbit anti-PV (1:1000, Abcam, #11427), anti-RFP (1:500, Rockland, #600-401-379), Alexa Fluor 405, 488, and 546 conjugated secondary antibodies (1:500, Invitrogen, #A-31553, #A-11001, #A-11003, and #A-11035).

## PV cell counting

To count PV+ interneurons, Z-stack images of immunostained striatal sections were taken on a confocal microscope (Olympus FLUOVIEW FV3000) with a 10 x or 20 x objective (Olympus # 1-U2B824 or Olympus # 1-U2B825) and step size of 2 μm. For quantification, image stacks were Z-projected to maximum intensity using Fiji (ImageJ) and cropped to a 400 μm x 400 μm image in anatomically matched sections of the DLS. All PV-expressing cells within this region were counted using the ROI manager tool in ImageJ. Designation of a cell as PV positive was determined by the experimenter and consistently maintained across animals. The experimenter was blind to genotype, and ROIs were made on the DAPI channel to avoid selecting regions based on PV expression. To quantify bulk PV fluorescence, ROIs were manually defined in ImageJ using the Freehand tool to cover as much of the DLS as possible, and mean fluorescence intensity was measured. To quantify individual cell PV fluorescence, ROIs were manually defined around every PV-positive cell in the previously drawn DLS ROI using the Freehand tool, and mean fluorescence intensity was measured.

## Western blot

Adult mice (P48-55) were deeply anesthetized with isoflurane and decapitated. Brains were rapidly dissected, and 1.5 mm dorsal striatum punches (Biopunch, Ted Pella, #15111–15) were collected from both hemispheres, flash-frozen in liquid nitrogen, and stored at −80 ° C. On the day of analysis, frozen samples were sonicated (QSonica Q55) until homogenized in 200 μl lysis buffer containing 1% SDS in 1 x PBS with Halt phosphatase inhibitor cocktail (Thermo Fisher Scientific, #PI78420) and Complete mini EDTA-free protease inhibitor cocktail (Roche, #4693159001). Sample homogenates were then boiled on a heat block at 95 ° C for 5 min and allowed to cool to RT. Total protein content was determined using a BCA assay (Thermo Fisher Scientific, #23227). Following the BCA assay, protein homogenates were mixed with 4 x Laemmli sample buffer (Bio-Rad, #161–0747). 12.5 μg of total protein per sample were then loaded onto 12% Criterion TGX gels (Bio-Rad, #5671044) and run at 65 V. Proteins were transferred to a PVDF membrane (Bio-Rad, #1620177) at 11 V for 14 hr at 4 ° C using the BioRad Criterion Blotter (Bio-Rad, #1704070). Membranes (Bio-Rad, #1620177) were briefly reactivated in methanol and rinsed in water 3 x. After rinsing, membranes were blocked in 5% milk in 1 x TBS with 1% Tween (TBS-Tween) for 1 hr at RT before being incubated with primary antibodies diluted in 5% milk in TBS-Tween overnight at 4 ° C. The following day, after 3x10 min washes with TBS-Tween, membranes were incubated with secondary antibodies for 1 hr at RT. Following 6×10 min washes, membranes were incubated with chemiluminescence substrate (PerkinElmer #NEL105001EA) for 1 min and exposed to Amersham Hyperfilm ECL (VWR, #95017–661).

Bands were quantified by densitometry using ImageJ software. GAPDH was used to normalize protein content, and data are expressed as a percentage of control within a given experiment. The following antibodies were used: anti-Caspr2 (1:5000, Abcam, #153856), anti-PV (1:2500, Abcam, #11427), anti-GAPDH (1:5000, Cell Signaling, #51745 S), and anti-rabbit goat HRP conjugate (1:5000, BioRad, #1705046).

## In situ hybridization

Fluorescent in situ hybridization was performed to quantify *Pvalb* mRNA expression in the striatum of *Cntnap2*$^{+/+}$ and *Cntnap2*$^{-/-}$ mice. Mice were briefly anesthetized with isoflurane, and brains were harvested, flash-frozen in OCT mounting medium (Thermo Fisher Scientific, #23-730-571) on dry ice and stored at –80 ° C for up to 6 months. 16 µm sections were collected using a cryostat (Thermo Fisher Scientific, Microm HM 550), mounted directly onto Superfrost Plus glass slides (VWR, #48311–703) and stored at –80 ° C for up to 6 months. In situ hybridization was performed according to the protocols provided with the RNAscope Multiplex Fluorescent Reagent Kit (ACD, #323100). *Drd1a* mRNA was visualized with a probe in channel 2 (ACD, #406491-C2) and *Pvalb* mRNA in channel 3 (ACD, #421931-C3). After incubation, sections were secured on slides using ProLong Gold Antifade Mountant with DAPI (Invitrogen, P36935) and 60x24 mm rectangular glass coverslips (VWR, #16004–096). Sections were imaged on an Olympus FluoView 3000 confocal microscope using a 10 x objective with 1.5 x zoom and a step size of 2 µm. *Pvalb*-expressing cells were quantified across the entire striatum using the ROI manager tool in ImageJ. A cell was considered *Pvalb* positive if over 50% of the cell contained fluorescent puncta when compared to the DAPI channel. The experimenter was blind to genotype.

## Behavioral analysis

All behavior studies were carried out in the dark phase of the light cycle under red lights (open field) or white lights (marble burying, holeboard, rotarod, and four choice reversal learning). Mice were habituated to the behavior testing room for at least 30 min prior to testing. Mice were given at least one day between different tests. All behavior equipment was cleaned between each trial and mouse with 70% ethanol and rinsed in diluted soap followed by water at the end of the day. If male and female mice were to be tested on the same day, male mice were run first then returned to the housing room, after which all equipment was thoroughly cleaned prior to bringing in female mice for habituation. Behavioral tests were performed with young adult male and female mice (7–11 weeks old). The experimenter was blind to genotype throughout the testing and scoring procedures.

## Open field assay

Exploratory behavior in a novel environment and general locomotor activity were assessed by a 60 min session in an open field chamber (40 cm L x 40 cm W x 34 cm H) made of transparent plexiglass. Horizontal infrared photobeams (Stoelting, 60001–02 A) were positioned to detect rearing. The mouse was placed in the bottom right-hand corner of the arena, and behavior was recorded using an overhead camera and analyzed using ANY-maze software (Stoelting). An observer manually scored self-grooming behavior during the first 20 min of the test. A grooming bout was defined as an unbroken series of grooming movements, including licking of body, paws, or tail, as well as licking of forepaws followed by rubbing of face with paws.

## Open field assay with DeepLabCut Keypoint-MoSeq analysis

Mice were placed in the open field arena and video recorded with a monochrome camera (FLIR Grasshopper 3, GS3-U3-41C6NIR-C) and a 16 mm wide angle lens (Kowa, LM16HC) placed above the arena from a height of 50 cm. To extract the body part (keypoint) coordinates from the video recordings, DeepLabCut (DLC) 2.3.4 (*Mathis et al., 2018*; *Nath et al., 2019*) was used. Fourteen body parts including nose, head, left ear, right ear, left forelimb, right forelimb, spine 1, spine 2, spine 3, left hindlimb, right hindlimb, tail 1, tail 2, and tail 3 were manually labeled on a small subset of the video frames. A DLC model was then trained using the annotated frames to label those 14 body parts for all videos recorded. The total distance traveled and the number of center entries were calculated using the coordinate of body part tail 1. Discrete behavior syllables were extracted using Keypoint-MoSeq 0.4.4 (*Weinreb et al., 2024*). Syllable usage and transition data were obtained using built-in functions of the Keypoint-MoSeq package. Decoding and entropy analysis were performed using a customized Python 3.9 script. Code available in GitHub (copy archived at *Wang and BateupLab, 2025*). Entropy was calculated using the following equation, where $u_i$ denotes the frequency of the syllable $i$ and $p_{i,j}$ denotes the transition probability from syllable $i$ to syllable $j$.: $Entropy = -\sum_{i,j} u_i \times p_{i,j} \times \log_2 p_{i,j}$

## Marble burying assay

The marble burying assay was used to test for repetitive behavior. 20 black marbles were arranged in an orderly 4x5 grid on top of 5 cm of clean corn cob bedding in a standard mouse cage. Overhead

room lights were on, and white noise was played to induce mild stress. Mice were placed in the cage with the marbles for 30 min. The number of unburied marbles (>50% exposed) was recorded after the session.

## Holeboard assay

The holeboard assay was used to measure exploratory and repetitive behavior. The holeboard apparatus consisted of a smooth, flat, opaque gray plastic platform, suspended 10 cm from the base by four plastic pegs in each corner. The board contained 16 evenly spaced 2 cm diameter holes and was surrounded by a 30 cm high clear plastic square encasing. During testing, mice were placed into the center of the holeboard. Mice explored the board for 10 min while video was recorded from both an above and side-view camera. Videos were used post-hoc to manually count and map the number of nose pokes made during the task. Nose pokes were defined as the mouse's nose passing through the board barrier when viewed through the side-view camera.

## Accelerating rotarod assay

The accelerating rotarod test was used to examine motor coordination and learning. Mice were trained on a rotarod apparatus (Ugo Basile, #47650) for four consecutive days. Three trials were completed per day with a 5 min break between trials. The rotarod was accelerated from 5 to 40 revolutions per minute (rpm) over 300 s for trials 1–6 (days 1 and 2), and from 10 to 80 rpm over 300 s for trials 7–12 (days 3 and 4). On the first testing day, mice were first acclimated to the apparatus by being placed on the rotarod rotating at a constant 5 rpm for 60 s and returned to their home cage for 5 min prior to starting trial 1. Latency to fall, or to rotate off the top of the rotarod barrel, was measured by the rotarod stop-trigger timer.

## Four choice odor-based reversal learning test

The four-choice odor-based reversal learning test was used to assess learning and cognitive flexibility. Animals were food restricted for 6 days in total, with unrestricted access to drinking water, and maintained at 90–95% of ad lib feeding body weight. Food was given at the end of the day once testing was completed. Food restriction and introduction to Froot Loop cereal pieces (Kellogg's, Battle Creek, MI) began 48 hr before pre-training. The four-choice test was performed in a custom-made square box (30.5 cm L×30.5 cm W × 23 cm H) constructed of clear acrylic. Four internal walls 7.6 cm wide partially divided the arena into four quadrants. A 15.2 cm diameter removable cylinder fit in the center of the maze and was lowered between trials (after a digging response) to isolate the mouse from the rest of the maze. Odor stimuli were presented mixed with wood shavings in white ceramic pots measuring 7.3 cm in diameter and 4.5 cm deep. All pots were sham baited with a piece of Froot Loop cereal secured underneath a mesh screen at the bottom of the pot. This was to prevent mice from using the odor of the Froot Loop to guide their choice. The apparatus was cleaned with 2.5% acetic acid followed by water, and the pots were cleaned with 70% ethanol followed by water between mice. The apparatus was cleaned with diluted soap and water at the end of each testing day.

On the first habituation day of pre-training (day 1), animals were allowed to freely explore the testing arena for 30 min and consume small pieces of Froot Loops placed inside empty pots positioned in each of the four corners. On the second shaping day of pre-training (day 2), mice learned to dig to find cereal pieces buried in unscented coarse pine wood shavings (Harts Mountain Corporation, Secaucus, NJ). A single pot was used, and increasing amounts of unscented wood shavings were used to cover each subsequent cereal reward. The quadrant containing the pot was alternated on each trial, and all quadrants were rewarded equally. Trials were untimed and consisted of (in order): two trials with no shavings, two trials with a dusting of shavings, two trials with the pot a quarter full, two trials with the pot half full, and four trials with the cereal piece completely buried by shavings. The mouse was manually returned to the center cylinder between trials.

On the days for odor discrimination (day 3, acquisition) and reversal (day 4), wood shavings were freshly scented on the day of testing. Anise extract (McCormick, Hunt Valley, MD) was used undiluted at 0.02 ml/g of shavings. Clove, litsea, and eucalyptus oils (San Francisco Massage Supply Co., San Francisco, CA) were diluted 1:10 in mineral oil and mixed at 0.02 ml/g of shavings. Thymol (thyme; Alfa Aesar, A14563) was diluted 1:20 in 50% ethanol and mixed at 0.01 ml/g of shavings. During the discrimination phase (day 3), mice had to discriminate between four pots with four different odors

and learn which one contained a buried food reward. Each trial began with the mouse confined to the start cylinder. Once the cylinder was lifted, timing began, and the mouse could freely explore the arena until it chose to dig in a pot. Digging was defined as purposefully moving the shavings with both front paws. A trial was terminated if no choice was made within 3 min and recorded as omission. If a mouse had three pairs of omissions, they were removed to their home cage for a 15–20 min break. After the break, if the mouse had three additional pairs of omissions then the task was terminated and the mouse was excluded from the dataset. Similarly, if the mouse took longer than 3 hr on the reversal without varying its response behavior, then it was also excluded from the dataset. Criterion was met when the animal completed eight out of ten consecutive trials correctly. The spatial location of the odors was shuffled on each trial. The rewarded odor during acquisition was anise.

The first four odor choices made during acquisition were analyzed to determine innate odor preference by the percentage of choices for a given odor: $Cntnap2^{+/+}$ mice: 60% thyme, 25% anise, 12.5% clove, and 2.5% litsea. $Cntnap2^{-/-}$ mice: 47.5% thyme, 45% anise, 7.5% clove, 0% litsea. We note that both $Cntnap2^{+/+}$ and $Cntnap2^{-/-}$ mice exhibited the strongest innate preference for thyme, an unrewarded odor. There were no significant differences in innate odor preference.

The reversal phase of the task was carried out on day 4. Mice first performed the task with the same rewarded odor as the discrimination day to ensure they learned and remembered the task. After reaching the criterion on recall (eight out of ten consecutive trials correct), the rewarded odor was switched, and mice underwent a reversal learning test in which a previously unrewarded odor (clove) was rewarded. A novel odor (eucalyptus) was also introduced, which replaced thyme. Perseverative errors were choices to dig in the previously rewarded odor that was no longer rewarded. Regressive errors were choosing the previously rewarded odor after the first correct choice of the newly rewarded odor. Novel errors were choices to dig in the pot with the newly introduced odor (eucalyptus). Irrelevant errors were choices to dig in the pot that had never been rewarded (litsea). Omissions were trials in which the mouse failed to make a digging choice within 3 min from the start of the trial. Total errors were the sum of perseverative, regressive, irrelevant, novel, and omission errors. Criterion was met when the mouse completed eight out of ten consecutive trials correctly. The spatial location of the odors was shuffled on each trial.

## Quantification and statistical analysis

Experiments were designed to compare the main effect of genotype. The sample sizes were based on prior studies and are indicated in the figure legend for each experiment. Whenever possible, quantification and analyses were performed blind to genotype. GraphPad Prism version 10 was used to perform statistical analyses. The statistical tests and outcomes for each experiment are indicated in the respective figure legend. Two-tailed unpaired t tests were used for comparisons between two groups. For data that did not pass the D'Agostino & Pearson normality test, a Mann-Whitney test was used. Two-way ANOVAs or mixed effects models were used to compare differences between groups for experiments with two independent variables. Statistical significance was defined in the figure panels as follows: *p<0.05, **p<0.01, ***p<0.001.

## Acknowledgements

This work was supported by Simons Foundation Autism Research Initiative (SFARI) research grant #514428 to HSB and NIH fellowship #F31NS124499 to KRC.

## Additional information

### Funding

| Funder | Grant reference number | Author |
|---|---|---|
| National Institute of Neurological Disorders and Stroke | F31NS124499 | Katherine R Cording |

| Funder | Grant reference number | Author |
|--------|------------------------|--------|
| Simons Foundation Autism Research Initiative | 514428 | Helen S Bateup |

The funders had no role in study design, data collection and interpretation, or the decision to submit the work for publication.

## Author contributions

Katherine R Cording, Formal analysis, Funding acquisition, Investigation, Visualization, Methodology, Writing – original draft, Writing – review and editing; Emilie M Tu, Formal analysis, Investigation, Methodology, Writing – review and editing; Hongli Wang, Software, Formal analysis, Methodology, Writing – review and editing; Alexander HCW Agopyan-Miu, Investigation; Helen S Bateup, Conceptualization, Supervision, Funding acquisition, Project administration, Writing – review and editing

## Author ORCIDs

Katherine R Cording ⬡ https://orcid.org/0000-0002-9921-5161
Emilie M Tu ⬡ https://orcid.org/0009-0001-6991-6805
Alexander HCW Agopyan-Miu ⬡ https://orcid.org/0000-0003-1914-7857
Helen S Bateup ⬡ https://orcid.org/0000-0002-0135-0972

## Ethics

All animal procedures were conducted in accordance with protocols approved by the University of California, Berkeley Institutional Animal Care and Use Committee (IACUC) and Office of Laboratory Animal Care (OLAC), protocol #AUP-2016-04-8684-3.

Reviewer #1 (Public review): https://doi.org/10.7554/eLife.100162.3.sa1
Reviewer #2 (Public review): https://doi.org/10.7554/eLife.100162.3.sa2
Reviewer #3 (Public review): https://doi.org/10.7554/eLife.100162.3.sa3
Author response https://doi.org/10.7554/eLife.100162.3.sa4

# Additional files

## Supplementary files

Supplementary file 1. Summary of behavior data by sex and genotype. Table reporting the behavior test results for each assay, separated by sex and genotype.

Source data 1. Source data for all figures, broken down by figure panel. Excel spreadsheet containing the numerical data used to create all figures, broken down into tabs for each figure panel.

MDAR checklist

## Data availability

All data generated or analysed during this study are included in the manuscript and supporting files. *Source data 1* contains all the numerical data used to generate the figures. Other raw data associated with this article are available on Dryad. The DeepLabCut Keypoint-MoSeq analysis of the open field data and code can be found at GitHub (copy archived at *Wang and BateupLab, 2025*).

The following dataset was generated:

| Author(s) | Year | Dataset title | Dataset URL | Database and Identifier |
|-----------|------|---------------|-------------|-------------------------|
| Cording KR, Tu EM, Wang H, Agopyan-Miu AHCW, Bateup HS | 2025 | Data from: Cntnap2 loss drives striatal neuron hyperexcitability and behavioral inflexibility | https://doi.org/10.5061/dryad.5x69p8dh7 | Dryad Digital Repository, 10.5061/dryad.5x69p8dh7 |

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
