## [Editor Report · eLife Assessment]

This **important** and well-executed study describes how deleting the autism spectrum disorder risk gene CNTNAP2 in mice increases dorsolateral striatal projection neuron excitability and promotes repetitive behaviors and cognitive inflexibility. The evidence supporting this claim is **convincing**. The study provides a potential cellular explanation for the repetitive and inflexible behavior in Cntnap2 knockout mice and CNTNAP2 disorder in humans, which would interest both basic and translational neuroscientists.

---

## [Referee Report · Reviewer #1 (Public review)]

Summary:

Cording et al. investigated how deletion of CNTNAP2, a gene associated with autism spectrum disorder, alters corticostriatal engagement and behavior. Specifically, the authors present slice electrophysiology data showing that striatal projection neurons (SPNs) are more readily driven to fire action potentials in response to stimulation of corticostriatal afferents, and this is due to increases in SPN intrinsic excitability rather than changes in excitatory or inhibitory synaptic inputs. Specifically, these changes seem to be due to preferential reduction of Kv1.2 in dSPNs. The authors separately show that CNTNAP2 mice display repetitive behaviors, enhanced motor learning and cognitive inflexibility. Overall, the authors' conclusions are supported by their data, but a few claims could use some more evidence to be convincing.

Strengths:

The use of multiple behavioral techniques, both traditional and cutting-edge machine learning-based analyses, provides a powerful means of assessing repetitive behaviors and behavioral transitions/rigidity. Characterization of both excitatory and inhibitory synaptic responses in slice electrophysiology experiments offers a broad survey of the synaptic alterations that may lead to increased corticostriatal engagement of SPNs.

Weaknesses:

As it stands, the reported changes in dorsolateral striatum SPN excitability are only correlative with reported changes in repetitive behaviors, motor learning and cognitive flexibility. The authors do broach this in the text (particularly in "Limitations and future directions").

---

## [Referee Report · Reviewer #2 (Public review)]

Summary:

This is an important study characterizing striatal dysfunction and behavioral deficits in Cntnap2-/- mice. There is growing evidence suggesting that striatal dysfunction underlies core symptoms of ASD but the specific cellular and circuit level abnormalities disrupted by different risk genes remain unclear. This study addresses how deletion of Cntnap2 affects the intrinsic properties and synaptic connectivity of striatal spiny projection neurons (SPN) of the direct (dSPN) and indirect (iSPN) pathways. Using Thy1-ChR2 mice and optogenetics the authors found increased firing of both types of SPNs in response to cortical afferent stimulation. However, there was no significant difference in the amplitude of optically-evoked excitatory postsynaptic currents (EPSCs) or spine density between Cntnap2-/- and WT SPNs, suggesting that the increased corticostriatal coupling might be due to changes in intrinsic excitability. Indeed, the authors found Cntnap2-/- SPNs, particularly dSPNs, exhibited higher intrinsic excitability, reduced rheobase current and increased membrane resistance compared to WT SPNs. The enhanced spiking probability in Cntnap2-/- SPNs is not due to reduced inhibition. Despite previous reports of decreased parvalbumin-expressing (PV) interneurons in various brain regions of Cntnap2-/- mice, the number and function (IPSC amplitude and intrinsic excitability) of these interneurons in the striatum were comparable to WT controls.

This study also includes a comprehensive behavioral analysis of striatal related behaviors. Cntnap2-/- mice demonstrated increased repetitive behaviors (RRBs), including more grooming bouts, increased marble burying, and increased nose poking in the holeboard assay. MoSeq analysis of behavior further showed signs of altered grooming behaviors and sequencing of behavioral syllables. Cntnap2-/- mice also displayed cognitive inflexibility in a four-choice odor-based reversal learning assay. While they performed similarly to WT controls during acquisition and recall phases, they required significantly more trials to learn a new odor-reward association during reversal, consistent with potential deficits in corticostriatal function.

Strengths:

This study provides significant contributions to the field. The finding of altered SPN excitability, the detailed characterization of striatal inhibition, and the comprehensive behavioral analysis are novel and valuable to understand the pathophysiology of Cntnap2-/- mice.

Weaknesses:

All my concerns were addressed in the revised version of the manuscript

---

## [Referee Report · Reviewer #3 (Public review)]

Summary:

The authors analyzed Cntnap2 KO mice to determine whether loss of the ASD risk gene CNTNAP2 alters the dorsal striatum's function.

Strengths:

The results demonstrate that loss of Cntnap2 results in increased excitability of striatal projection neurons (SPNs) and altered striatal-dependent behaviors, such as repetitive, inflexible behaviors. Unlike other brain areas and cell types, synaptic inputs onto SPNs were normal in Cntnap2 KO mice. The experiments are well-designed, and the results support the authors' conclusions.

Weaknesses:

The mechanism underlying SPN hyperexcitability was not explored, and it is unclear whether this cellular phenotype alone can account for the behavioral alterations in Cntnap2 KO mice. No clear explanation emerges for the variable phenotype in different brain areas and cell types.

Comments on revisions:

The authors have appropriately addressed all my comments. In my opinion, no further changes are required.

---

## [Author Response]

The following is the authors’ response to the original reviews

We thank the reviewers for their careful review of our manuscript and the constructive comments. We have addressed the majority of comments with either new experiments, analyses, and/or text revisions. A summary of the major changes is listed below, followed by our point-by-point responses to the reviewer comments.

Major changes:

(1) We sought to gain insight into the potential mechanistic cause of the increased intrinsic excitability of *Cntnap2^-/-^* dSPNs. Given that Kv1.1 and 1.2 potassium channels are known to interact with Caspr2 (the protein encoded by *Cntnap2*), we hypothesized that altered number, location, and/or function of these channels may underlie the excitability change in these cells. To investigate this, we performed new analyses of the initial dataset to assess action potential (AP) properties known to be impacted by potassium channel function. Indeed, we found that AP frequency was increased, and rheobase current, AP latency and AP threshold were decreased in *Cntnap2^-/-^* dSPNs, suggestive of altered Kv1.2 function. These data are in the new Figure 3—figure supplement 1. We also performed new electrophysiology experiments in which we pharmacologically blocked Kv1.1 and 1.2 to assess whether the effects of blocking these channels would be occluded in *Cntnap2^-/-^* dSPNs. We found that (1) WT dSPNs responded to blockade of Kv1.1/1.2 channels by increasing their excitability but *Cntnap2^-/-^* dSPNs did not and (2) Kv1.1/1.2 channels were more important contributors to the excitability of dSPNs compared to iSPNs. These new data are presented in the revised Figure 4 and Figure 4—figure supplement 1 and Figure 4—figure supplement 2.

(2) We performed additional experiments to assess excitatory synaptic properties, specifically AMPA/NMDA receptor ratio. This has been added to Figure 1.

(3) We performed more rigorous statistical analyses of the initial physiology datasets to align with the statistics performed for the revision experiments. This applies to Figure 1, Figure 2, Figure 3, Figure 5, and Figure 2—figure supplement 1.

(4) In the discussion section, we now highlight potential limitations of the study and further discuss the variable impact that *Cntnap2* loss has on different cell types and brain regions.

**Reviewer #1 (Public Review):**
Summary:Cording et al. investigated how deletion of CNTNAP2, a gene associated with autism spectrum disorder, alters corticostriatal engagement and behavior. Specifically, the authors present slice electrophysiology data showing that striatal projection neurons (SPNs) are more readily driven to fire action potentials in response to stimulation of corticostriatal afferents, and this is due to increases in SPN intrinsic excitability rather than changes in excitatory or inhibitory synaptic inputs. The authors show that CNTNAP2 mice display repetitive behaviors, enhanced motor learning, and cognitive inflexibility. Overall the authors' conclusions are supported by their data, but a few claims could use some more evidence to be convincing.Strengths:The use of multiple behavioral techniques, both traditional and cutting-edge machine learning-based analyses, provides a powerful means of assessing repetitive behaviors and behavioral transitions/rigidity.Characterization of both excitatory and inhibitory synaptic responses in slice electrophysiology experiments offers a broad survey of the synaptic alterations that may lead to increased corticostriatal engagement of SPNs.Weaknesses:(1) The authors conclude that increased cortical engagement of SPNs is due to changes in SPN intrinsic excitability rather than synaptic strength (either excitatory or inhibitory). One weakness is that only AMPA receptor-mediated responses were measured. Though the holding potential used for experiments in Figure 1FI wasn't clear, recordings were presumably performed at a hyperpolarized potential that limits NMDA receptormediated responses. Because the input-output experiments used to conclude that corticostriatal engagement of SPNs is elevated (Figure 1B-E) were conducted in the current clamp, it is possible that enhanced NMDA receptor engagement contributed to increased SPN responses to cortical stimulation. Confirming that NMDA receptor-mediated EPSC components are not altered would strengthen the main conclusion.

The reviewer is correct, the initial optically-evoked EPSC assessments were performed at a hyperpolarized potential (-70mV), thus measuring primarily AMPAR-mediated currents. We agree that assessing potential changes in the NMDAR-mediated EPSC component is important and we have completed new experiments to assess this. We find no differences in NMDAR-mediated EPSCs assessed at +40mV or the AMPA:NMDA ratio.

These results have been added to Figure 1. An expanded analysis of these results is shown in Author response image 1. We note that the previous AMPAR-mediated EPSC results have been replicated in this additional experiment, again showing no change in *Cntnap2^-/-^* SPNs.

**Author response image 1. sa4fig1:** AMPA and NMDA receptor-mediated EPSCs are unchanged in *Cntnap2^-/-^* SPNs. (A) Quantification (mean ± SEM) of AMPA:NMDA ratio per cell for *Cntnap2^+/+^* and *Cntnap2^-/-^* dSPNs, p=0.9537, Mann-Whitney test. (B) dSPN AMPA current per cell, p=0.6172, Mann-Whitney test. (C) dSPN NMDA current per cell, p=0.6009, Mann-Whitney test. (D) dSPN AMPA:NMDA ratio averaged by animal, p=0.8413, Mann-Whitney test. (E) dSPN AMPA current averaged by animal, p>0.9999, Mann-Whitney test. (F) dSPN NMDA current averaged by animal, p=0.6905, Mann-Whitney test. (G) Quantification (mean ± SEM) of AMPA:NMDA ratio per cell for *Cntnap2^+/+^* and *Cntnap2^-/-^* iSPNs, p=0.4104, Mann-Whitney test. (H) iSPN AMPA current per cell, p=0.9010, Mann-Whitney test. (I) iSPN NMDA current per cell, p=0.9512, two-tailed unpaired t test. (J) iSPN AMPA:NMDA averaged by animal, p=0.3095, Mann-Whitney test. (K) iSPN AMPA current averaged by animal, p=>0.9999, Mann-Whitney test. (L) iSPN NMDA current averaged by animal, p=0.8413, Mann-Whitney test. All values were recorded using 20% blue light intensity. For dSPNs: *Cntnap2^+/+^* n=22 cells from 5 mice, *Cntnap2^-/-^* n=22 cells from 5 mice. For iSPNs: *Cntnap2^+/+^* n=21 cells from 5 mice, *Cntnap2^-/-^* n=21 cells from 5 mice.

(2) Data clearly show that SPN intrinsic excitability is increased in knockout mice. Given that CNTNAP2 has been linked to potassium channel regulation, it would be helpful to show and quantify additional related electrophysiology data such as negative IV curve responses and action potential hyperpolarization.

We appreciate this suggestion. As indicated by the reviewer, Caspr2, has been shown to control the clustering of Kv1-family potassium channels in axons isolated from optic nerve and corpus callosum (PMIDs: 10624965, 12963709, 29300891). In particular, Caspr2 is known to associate directly with Kv1.2 (PMID: 29300891). To assess a potential contribution of Kv1.2 to the excitability phenotype, we performed additional analyses of our original dataset to quantify AP properties known to be impacted by changes in Kv1.2 function (i.e. latency to fire and AP threshold, new Figure 3—figure supplement 1). We identified several changes in *Cntnap2^-/-^* dSPNs resembling those that occur in wild-type cells when Kv1.2 is blocked (i.e. reduced threshold and reduced latency to fire, Figure 3—figure supplement 1).

We then performed a pharmacological experiment, blocking Kv1.1/1.2 using α-dendrotoxin (α-DTX) while recording intrinsic excitability to assess whether the effects of this drug on dSPN excitability were occluded in *Cntnap2^-/-^* cells. Indeed, we found that while blocking Kv1.1/1.2 in wild-type dSPNs significantly reduced threshold and increased intrinsic excitability, these effects were not seen in *Cntnap2^-/-^* dSPNs (new Figure 4). We believe that this suggests an altered contribution of Kv1.1/1.2 to the intrinsic excitability of mutant dSPNs, owing to a change in the clustering, number, or function of these channels. Therefore, loss-of-function of Kv1.1/1.2 is a likely explanation for the enhanced intrinsic excitability of *Cntnap2^-/-^* dSPNs. Interestingly, we found that α-DTX had only subtle effects on iSPNs (*Cntnap2* WT or mutant), suggesting a lesser contribution of these channel in controlling the excitability of indirect pathway cells. This finding can account for the relatively stronger effect of *Cntnap2* loss on dSPN physiology. The results of these new experiments and analyses are presented in the new Figure 4, Figure 4—figure supplement 1 and Figure 4—figure supplement 2.

(3) As it stands, the reported changes in dorsolateral striatum SPN excitability are only correlative with reported changes in repetitive behaviors, motor learning, and cognitive flexibility.

We agree that we have not identified a causative relationship between the change in dorsolateral dSPN excitability and the behaviors that we measured in *Cntnap2^-/-^* mice. That said, in a previous study, we showed that selective deletion of the autism spectrum disorder (ASD) risk gene *Tsc1* from dorsal striatal dSPNs increased corticostriatal drive and enhanced rotarod motor learning (PMID: 34380034). Therefore, while we have not demonstrated causality in this study, we hypothesize that changes in dSPN excitability are likely to contribute to the behavioral phenotypes observed in *Cntnap2^-/-^* mice.

**Reviewer #2 (Public Review):**
Summary:This is an important study characterizing striatal dysfunction and behavioral deficits in *Cntnap2-/-* mice. There is growing evidence suggesting that striatal dysfunction underlies core symptoms of ASD but the specific cellular and circuit level abnormalities disrupted by different risk genes remain unclear. This study addresses how the deletion of Cntnap2 affects the intrinsic properties and synaptic connectivity of striatal spiny projection neurons (SPN) of the direct (dSPN) and indirect (iSPN) pathways. Using Thy1-ChR2 mice and optogenetics the authors found increased firing of both types of SPNs in response to cortical afferent stimulation. However, there was no significant difference in the amplitude of optically-evoked excitatory postsynaptic currents (EPSCs) or spine density between *Cntnap2-/-* and WT SPNs, suggesting that the increased corticostriatal coupling might be due to changes in intrinsic excitability. Indeed, the authors found *Cntnap2-/-* SPNs, particularly dSPNs, exhibited higher intrinsic excitability, reduced rheobase current, and increased membrane resistance compared to WT SPNs. The enhanced spiking probability in *Cntnap2-/-* SPNs is not due to reduced inhibition. Despite previous reports of decreased parvalbumin-expressing (PV) interneurons in various brain regions of *Cntnap2-/-* mice, the number and function (IPSC amplitude and intrinsic excitability) of these interneurons in the striatum were comparable to WT controls.This study also includes a comprehensive behavioral analysis of striatal-related behaviors. *Cntnap2-/-* mice demonstrated increased repetitive behaviors (RRBs), including more grooming bouts, increased marble burying, and increased nose poking in the holeboard assay. MoSeq analysis of behavior further showed signs of altered grooming behaviors and sequencing of behavioral syllables. *Cntnap2-/-* mice also displayed cognitive inflexibility in a four-choice odor-based reversal learning assay. While they performed similarly to WT controls during acquisition and recall phases, they required significantly more trials to learn a new odor-reward association during reversal, consistent with potential deficits in corticostriatal function.Strengths:This study provides significant contributions to the field. The finding of altered SPN excitability, the detailed characterization of striatal inhibition, and the comprehensive behavioral analysis are novel and valuable to understanding the pathophysiology of *Cntnap2-/-* mice.Weaknesses:(1) The approach based on Thy-ChR2 mice has the advantage of overcoming issues caused by injection efficiency and targeting variability. However, the spread of oEPSC amplitudes across mice shown in panels of Figure 1 G/I is very high with almost one order of magnitude difference between some mice. Given this is one of the most important points of the study it will be important to further analyze and discuss what this variability might be due to. Typically, in acute slice recordings, the within-animal variability is larger than the variability across animals. From the sample sizes reported it seems the authors sampled a large number of animals, but with a relatively low number of neurons per animal (per condition). Could this be one of the reasons for this variability?

We agree with the reviewer that the variability in these experiments is quite large. We have replicated these experiments in the process of performing AMPA:NMDA ratio recordings (see above response to Reviewer 1’s comment). We again find no differences in AMPAR-mediated EPSC amplitude between WT and mutant SPNs (Author response image 2). Notably, these experiments also demonstrate a large amount of variability. In the original dataset, a small number of cells were collected from each animal (~1-3 cells/mouse). However, the variability remains in the new dataset, in which more cells were collected from each animal (~4-6 cells/mouse). We find both within-animal and between-animal variability, as can be seen in Author response image 2 (recordings made from the same animal are color-coordinated). Potential sources of variability in this experiment include: (1) variable expression of ChR2 per mouse, (2) variable innervation of ChR2-expressing terminals onto any given recorded cell, and/or (3) differences in prior plasticity state between cells (i.e. some neurons may have recently undergone corticostriatal LTP or LTD).

**Author response image 2. sa4fig2:** Optically-evoked AMPAR EPSCs exhibit within- and between-animal variability. (A) Quantification of EPSC amplitude evoked in dSPNs at different light intensities from the original dataset, plotted by cell (line represents the mean, dots/squares represent average EPSC amplitude for each recorded cell). *Cntnap2^+/+^* n=17 cells from 8 mice, *Cntnap2^-/-^* n=13 cells from 5 mice. Repeated measures two-way ANOVA p values are shown; g x s F (2, 56) = 0.3879, geno F (1, 28) = 0.8098, stim F (1.047, 29.32) = 76.56. (B) Quantification of EPSC amplitude evoked in dSPNs, averaged by mouse (line represents the mean, dots/squares represent average EPSC amplitude for each mouse). *Cntnap2^+/+^* n=8 mice, *Cntnap2^-/-^* n=5 mice. Repeated measures two-way ANOVA p values are shown; g x s F (2, 22) = 0.2154, geno F (1, 11) = 0.2585, stim F (1.053, 11.58) = 49.68. (C) Quantification of EPSC amplitude in dSPNs from the revision dataset, plotted by cell (line represents the mean, dots/squares represent average EPSC amplitude for each recorded cell). *Cntnap2^+/+^* n=22 cells from 5 mice, *Cntnap2^-/-^* n=22 cells from 5 mice. Repeated measures two-way ANOVA p values are shown; g x s F (2, 84) = 0.01885, geno F (1, 42) = 0.002732, stim F (1.863, 78.26) = 20.93. (D) Quantification of EPSC amplitude in dSPNs from the revision dataset, averaged by mouse (line represents the mean, dots/squares represent average EPSC amplitude for each mouse). *Cntnap2^+/+^* n=5 mice, *Cntnap2^-/-^* n=5 mice. Repeated measures two-way ANOVA p values are shown; g x s F (2, 16) = 0.06288, geno F (1, 8) = 0.006548, stim F (1.585, 12.68) = 16.97. (E) Quantification of EPSC amplitude evoked in iSPNs from the original dataset, plotted by cell (line represents the mean, dots/squares represent average EPSC amplitude for each recorded cell). *Cntnap2^+/+^* n=13 cells from 6 mice, *Cntnap2^-/-^* n=11 cells from 5 mice. Repeated measures two-way ANOVA p values are shown; g x s F (2, 44) = 0.9414, geno F (1, 22) = 1.333, stim F (1.099, 24.18) = 52.26. (F) Quantification of EPSC amplitude evoked in iSPNs from original dataset, averaged by mouse (line represents the mean, dots/squares represent average EPSC amplitude for each mouse). *Cntnap2^+/+^* n=6 mice, *Cntnap2^-/-^* n=5 mice. Repeated measures two-way ANOVA p values are shown; g x s F (2, 18) = 0.4428, geno F (1, 9) = 0.5635, stim F (1.095, 9.851) = 23.82. (G) Quantification of EPSC amplitude evoked in iSPNs from the revision dataset, plotted by cell (line represents the mean, dots/squares represent average EPSC amplitude for each recorded cell). *Cntnap2^+/+^* n=21 cells from 5 mice, *Cntnap2^-/-^* n=21 cells from 5 mice. Repeated measures two-way ANOVA p values are shown; g x s F (2, 80) = 0.04134, geno F (1, 40) = 0.007025, stim F (1.208, 48.31) = 102.9. (H) Quantification of EPSC amplitude evoked in iSPNs from the revision dataset, averaged by mouse (line represents the mean, dots/squares represent average EPSC amplitude for each mouse). *Cntnap2^+/+^* n=5 mice, *Cntnap2^-/-^* n=5 mice. Repeated measures two-way ANOVA p values are shown; g x s F (2, 16) = 0.001865, geno F (1, 8) = 0.1004, stim F (1.179, 9.433) = 61.31.

(2) This is particularly important because the analysis of corticostriatal evoked APs in panels C and E is performed on pooled data without considering the variability in evoked current amplitudes across animals shown in G and I. Were the neurons in panels C/E recorded from the same mice as shown in G/I? If so, it would be informative to regress AP firing data (say at 20% LED) to the average oEPSC amplitude recorded on those mice at the same light intensity. However, if the low number of neurons recorded per mouse is due to technical limitations, then increasing the sample size of these experiments would strengthen the study.

We appreciate this point; however, the evoked AP experiment and the evoked EPSC experiment were performed on different mice, so it is not possible to correlate the data across experiments. While the evoked AP experiments were performed using potassium-based internal, we used a cesium-based internal to measure AMPAR-mediated EPSCs to more accurately detect synaptic currents. We note that the evoked AP experiments share a similar amount of variability as the evoked EPSC experiments, again possibly owing to variable expression of channelrhodopsin per mouse, variable innervation of ChR2-positive terminals onto individual cells, and/or differences in prior plasticity status between cells.

(3) On a similar note, there is no discussion of why iSPNs also show increased corticostriatal evoked firing in Figure 1E, despite the difference in intrinsic excitability shown in Figure 3. This suggests other potential mechanisms that might underlie altered corticostriatal responses. Given the role of Caspr2 in clustering K channels in axons, altered presynaptic function or excitability could also contribute to this phenotype, but potential changes in PPR have not been explored in this study.

We have now performed more rigorous statistics on the data in Figure 1 (repeated measures two-way ANOVA) such that the difference in corticostriatal evoked firing in *Cntnap2^-/-^* iSPNs no longer reaches statistical significance. This is consistent with the modest but statistically non-significant effect of *Cntnap2* loss on iSPN intrinsic excitability. We agree with the reviewer that presynaptic alterations could potentially contribute to the changes in cortically-driven action potentials, especially as this experiment was performed without any synaptic blockers present, and *Cntnap2* is deleted from all cells. That said, if changes in presynaptic release probability accounted for the increased corticostriatal drive, we would expect to see differences in cortically-evoked EPSCs onto SPNs.

While we can’t rule out the possibility of pre-synaptic changes, a straightforward explanation for our findings is that loss or alteration of Kv1.2 channel function is responsible for the increased excitability of *Cntnap2^-/-^* dSPNs, resulting in enhanced spiking in response to cortical input. Given the fact that Kv1.2 channels appear less important for regulating iSPN excitability (see new Figure 4 and Figure 4—figure supplement 2), this can explain the greater impact of *Cntnap2* loss on dSPN physiology.

(4) Male and female SPNs have different intrinsic properties but the number and/or balance of M/F mice used for each experiment is not reported.

We agree that this is an important consideration. Author response table 1 provides the sex breakdown for the intrinsic excitability experiments. While we did not explicitly power the experiments to test for sex differences, Author response image 3 shows the data separated by sex and genotype for the intrinsic excitability experiments. Within genotype, we find no significant differences between males and females, except for *Cntnap2^-/-^* iSPNs which showed a significant interaction between sex and current step (Author response image 3F). Interestingly, while present in both sexes, the excitability shift of *Cntnap2^-/-^* dSPNs may be slightly more pronounced in females compared to males (Author response image 3C and D). However, this result would require further validation with a greater sample size.

**Author response table 1. sa4table1:** Numbers of male and female mice used for the intrinsic excitability experiments.

Cell Type	Genotype	Sex	Cells (Animals)
dSPN	*Cntnap2^+/+^*	Male	12 (4)
Female	8 (4)
*Cntnap2^-/-^*	Male	12 (4)
Female	11 (4)
iSPN	*Cntnap2^+/+^*	Male	10 (4)
Female	12 (4)
*Cntnap2^-/-^*	Male	12 (4)
Female	9 (4)

**Author response image 3. sa4fig3:** Enhanced excitability of *Cntnap2^-/-^* dSPNs is present in both males and females. (A) Quantification (mean ± SEM) of the number of APs evoked in dSPNs in *Cntnap2^+/+^* males and females at different current step amplitudes. *Cntnap2^+/+^* males n=12 cells from 4 mice, *Cntnap2^+/+^* females n=8 cells from 4 mice. Repeated measures two-way ANOVA p values are shown; s x c F (28, 560) = 0.8992, sex F (1, 20) = 0.3754, current F (1.279, 25.57) = 56.85. (B) Quantification (mean ± SEM) of the number of APs evoked in dSPNs in *Cntnap2^-/-^* males and females at different current step amplitudes. *Cntnap2^-/-^* males n=12 cells from 4 mice, *Cntnap2^-/-^* females n=11 cells from 4 mice. Repeated measures two-way ANOVA p values are shown; s x c F (28, 588) = 0.6752, sex F (1, 21) = 0.04534, current F (2.198, 46.15) = 78.89. (C) Quantification (mean ± SEM) of the number of APs evoked in dSPNs in *Cntnap2^+/+^* males and *Cntnap2^-/-^* males at different current step amplitudes. *Cntnap2^+/+^* males n=12 cells from 4 mice, *Cntnap2^-/-^* males n=12 cells from 4 mice. Repeated measures two-way ANOVA p values are shown; g x c F (28, 672) = 2.233, geno F (1, 24) = 3.746, current F (1.708, 40.98) = 79.82. (D) Quantification (mean ± SEM) of the number of APs evoked in dSPNs in *Cntnap2^+/+^* females and *Cntnap2^-/-^* females at different current step amplitudes. *Cntnap2^+/+^* females n=8 cells from 4 mice, *Cntnap2^-/-^* females n=11 cells from 4 mice. Repeated measures two-way ANOVA p values are shown; g x c F (28, 476) = 1.547, geno F (1, 17) = 5.912, current F (1.892, 32.17) = 58.76. (E) Quantification (mean ± SEM) of the number of APs evoked in iSPNs in *Cntnap2^+/+^* males and females at different current step amplitudes. *Cntnap2^+/+^* males n=10 cells from 4 mice, *Cntnap2^+/+^* females n=12 cells from 4 mice. Repeated measures two-way ANOVA p values are shown; s x c F (28, 560) = 1.236, sex F (1, 20) = 1.074, current F (2.217, 44.34) = 179.6. (F) Quantification (mean ± SEM) of the number of APs evoked in iSPNs in *Cntnap2^-/-^* males and females at different current step amplitudes. *Cntnap2^-/-^* males n=12 cells from 4 mice, *Cntnap2^-/-^* females n=9 cells from 4 mice. Repeated measures two-way ANOVA p values are shown; s x c F (28, 532) = 2.513, sex F (1, 19) = 2.639, current F (1.858, 35.31) = 152.5. (G) Quantification (mean ± SEM) of the number of APs evoked in iSPNs in *Cntnap2^+/+^* males and *Cntnap2^-/-^* males at different current step amplitudes. *Cntnap2^+/+^* males n=10 cells from 4 mice, *Cntnap2^-/-^* males n=12 cells from 4 mice. Repeated measures two-way ANOVA p values are shown; g x c F (28, 560) = 0.4723, geno F (1, 20) = 0.5675, current F (2.423, 48.47) = 301.7. (H) Quantification (mean ± SEM) of the number of APs evoked in iSPNs in *Cntnap2^+/+^* females and *Cntnap2^-/-^* females at different current step amplitudes. *Cntnap2^+/+^* females n=12 cells from 4 mice, *Cntnap2^-/-^* females n=9 cells from 4 mice. Repeated measures two-way ANOVA p values are shown; g x c F (28, 532) = 1.655, geno F (1, 19) = 0.2322, current F (2.081, 39.55) = 99.45.

(5) There is no mention of how membrane resistance was calculated, and no I/V plots are shown.

Passive properties were calculated from the average of five -5 mV, 100 ms long test pulse steps applied at the beginning of every experiment. Membrane resistance was calculated from the double exponential curve fit. This has now been added to the methods section.

(6) It would be interesting to see which behavior transitions most contribute to the decrease in entropy. Are these caused by repeated or perseverative grooming bouts? Or is this inflexibility also observed across other behaviors? The transition map in Figure S5 shows the overall number of syllables and transitions but not their sequence during behavior. Can this be analyzed by calculating the ratio of individual 𝑢𝑖 × 𝑝𝑖,𝑗 × log2 𝑝𝑖,𝑗 factors across genotypes?

We thank the reviewer for raising an insightful question. Here we use a finite state Markov chain model to describe the syllable transitions in animal behavior. To quantify the randomness in the system, we calculated the entropy of the Markov chain (see methods section). The reviewer suggested calculating the partial entropy of the transition matrix, which would allow us to estimate the contribution of a subset of states to the entropy of the whole system, given by the equation:\begin{document}$$\displaystyle  \operatorname{Entropy}(S)=-\sum_{i \in S} u_{i} \sum_{j \in S} p_{i, j} \log \log \left(p_{i, j}\right)$$\end{document}

The partial equation can indeed quantify the stochasticity, or “flexibility” in our context, of the sub-system containing only a subset of the behavior syllables. However, there are two main limitations to this approach:

(1) The partial entropy fails to account for the transitions connecting the subset with the rest of the states in the system

(2) The stationary distribution may not reflect the actual probabilities in the isolated sub-system *S*.

Consequently, the partial entropy cannot be directly interpreted as the fraction of contributions from specific syllable pairs or sub-system to the entropy of the whole system. To be more specific, while a significant difference between the same sub-system in WT and KO groups could indicate that the sub-system contributes significantly to the difference of overall entropy, a non-significant result does not mean that the sub-system does not contribute to overall entropy difference, as interactions between the sub-system and other not-considered states are not accounted for.

**Author response image 4. sa4fig4:** Grooming syllables contribute to some but not all differences in syllable transitions in *Cntnap2^-/-^* mice. We calculated the entropy of each syllable pair using 𝑢𝑖 × 𝑝𝑖,𝑗 × log2 𝑝𝑖,𝑗 for every syllable pair and every animal. We then statistically tested the difference between genotypes for each syllable pair using Mann-Whitney tests. This plot displays those adjusted p-values for each syllable pair between WT and *Cntnap2^-/-^* groups. The significant p-values suggest that the transitions to syllables 24 and 25 are different between genotypes (note that these correspond to grooming syllables, see Figure 5N). However, since the overall entropy is a summation of every pair, it is difficult to conclude that syllables 24 and 25 are the sole contributors to the different entropy we observed.

**Reviewer #3 (Public Review):**
Summary:The authors analyzed Cntnap2 KO mice to determine whether loss of the ASD risk gene CNTNAP2 alters the dorsal striatum's function.Strengths:The results demonstrate that loss of Cntnap2 results in increased excitability of striatal projection neurons (SPNs) and altered striatal-dependent behaviors, such as repetitive, inflexible behaviors. Unlike other brain areas and cell types, synaptic inputs onto SPNs were normal in Cntnap2 KO mice. The experiments are welldesigned, and the results support the authors' conclusions.Weaknesses:The mechanism underlying SPN hyperexcitability was not explored, and it is unclear whether this cellular phenotype alone can account for the behavioral alterations in Cntnap2 KO mice. No clear explanation emerges for the variable phenotype in different brain areas and cell types.

We agree that identifying the mechanism by which *Cntnap2* loss affects intrinsic excitability is interesting and important. We have added experiments to address this and conclude that the improper clustering, number, or function of Kv.1/1.2 channels in *Cntnap2^-/-^* dSPNs is likely responsible for their increased excitability. These channels are known to be clustered/organized in part by Caspr2 (PMIDs: 10624965, 12963709, 29300891), and Kv1.1/1.2 channels are known to play an important role in regulating excitability in SPNs (PMIDs: 13679409, 32075716). In the case of dSPNs, blocking these channels with α-DTX significantly increased the excitability of WT cells (as has been previously reported); however, this effect was occluded in mutant cells, perhaps owing to a decreased contribution of Kv1.1/1.2 channels to excitability in *Cntnap2^-/-^* dSPNs. In addition, we found that blockade of these channels with α-DTX only modestly affected the excitability of iSPNs. Therefore, this can explain why loss of *Cntnap2* more strongly affects the excitability of dSPNs. Please see new Figure 4, Figure 4—figure supplement 1 and Figure 4—figure supplement 2 for these new data.

We agree with the reviewer that we have not identified a causative relationship between the change in dSPN excitability and the behavioral alterations in *Cntnap2^-/-^* mice. This is a limitation of the study.

It is interesting to speculate on the root of the varying impacts to excitability that occur across different brain regions and cell types in *Cntnap2^-/-^* mice. Increased excitability, as we see in dSPNs, has been identified in cerebellar Purkinje cells and L2/3 pyramidal neurons in somatosensory cortex in the context of *Cntnap2* loss (PMIDs: 34593517, 30679017, 36793543). However, other cell types in *Cntnap2^-/-^* mice have exhibited no change in excitability (mPFC, L2/3 pyramidal neurons, PMID: 31141683) or hypoexcitability (subset of L5/6 pyramidal neurons, PMID: 29112191). While all of these cell types express Kv1.2 channels, they fundamentally vary in their intrinsic properties, owing to the role that other ion channels play in membrane excitability. As a result, loss of *Cntnap2* is expected to have a variable effect on excitability depending on the cell type and the complement of other ion channels that are present. In addition, an initial change in excitability may drive secondary, potentially compensatory, changes in other channels that lead to a different excitability state. These changes are also expected to be cell type-specific. We do note that both of the cell types that show increased excitability in the context of *Cntnap2* loss have been shown to exhibit an α-DTX-sensitive Kv1 channel current, such that application of α-DTX results in increased firing of these cells (cerebellar Purkinje cells; PMIDs: 17087603, 16210348 and L2/3 pyramidal neurons in somatosensory cortex; PMID: 17215507). These findings are consistent with our results in *Cntnap2^-/-^* dSPNs.

**Reviewer #1 (Recommendations For The Authors):**
More thorough analysis of some of the manually quantified behaviors would be helpful. For example, only the grooming bout number was presented- what about the duration of bouts and total time grooming? Similarly, for the open field the number of center entries was reported but what about the total time in the center?

We have quantified the time spent grooming and total time spent in the center during the open field test from our original data (Author response image 5). These data were not originally included in the manuscript because they were recorded for only a subset of the total animals. For each of these measures we find trend level changes, which are consistent with the primary measures reported in the main manuscript.

**Author response image 5. sa4fig5:** Time in the center of the open field and time spent grooming trend towards an increase in *Cntnap2^-/-^* mice. (A) Quantification (mean ± SEM) of total time spent in the center of the open field during a 60 minute test, p=0.0656, Mann-Whitney test. (B) Time spent self-grooming during the first 20 minutes of the open field test, p=0.0611, Mann-Whitney test. For both measurements, *Cntnap2^+/+^* n=18 mice, *Cntnap2^-/-^* n=19 mice.

**Reviewer #3 (Recommendations For The Authors):**
What accounts for the hyperexcitability observed in Cntnap2-deficient SPNs? The authors noted that excitability is reportedly increased, reduced, or unchanged in different brain areas. What accounts for this disparity? Is it about the subcellular localization of Kv1 channels? The authors may want to test this possibility experimentally. At least, they may want to test whether Kv1 channels are mislocalized in SPNs.

We agree that this is an important point, and we have performed additional experiments to address this. We find that the Kv1.1/1.2 blocker a-DTX significantly increases the excitability of WT dSPNs but not *Cntnap2^-/-^* dSPNs. This suggests that the mechanism underlying dSPN hyperexcitability in *Cntnap2* mutants is the improper clustering, number, or function of Kv1.1/1.2 channels. These channels are known to be clustered and organized in part by Caspr2 (PMIDs: 10624965, 12963709, 29300891) and have been shown to play an important role in regulating the excitability of SPNs (PMIDs: 13679409, 32075716). Interestingly, we find that a-DTX has less of an effect on the excitability of iSPNs, which may account for the greater impact of *Cntnap2* loss on dSPNs. Please see new Figure 4, Figure 4—figure supplement 1 and Figure 4—figure supplement 2 for these added data and analyses.

Please see above response to Reviewer #3 for our speculation on the variable impact of *Cntnap2* loss on different cell types and brain regions.

We agree with the reviewer that assessing potential differences in subcellular localization of Kv1 channels in our model would bolster the conclusion that these channels are mislocalized in the *Cntnap2^-/-^* striatum. We piloted these experiments using immunohistochemistry to stain for Kv1.1 and 1.2 but found that without very high-resolution imaging, it would be challenging to accurately quantify Kv1 puncta in a cell type-specific manner. We instead chose to investigate the functional contribution of Kv1 channels to the dSPN hyperexcitability phenotype through the a-DTX experiments outlined above. α-DTX strongly inhibits Kv1.2 channels, but also Kv1.1 channels to some extent (PMIDs: 12042352, 13679409). We find that the effects of a-DTX on SPN excitability are occluded in *Cntnap2^-/-^* dSPNs; therefore, we conclude that Kv1.2 (and possibly Kv1.1) channels have reduced function in these cells. Further work will be needed to determine if this is a result of channel mislocalization or another type of alteration.

The authors did not detect synaptic changes in Cntnap-deficient SPNs. This important observation should be briefly discussed in the context of previous work in other brain regions and cell types. For example, some studies reported structural and functional changes at excitatory synapses. The variable impact on synapses suggests distinct compensatory mechanisms in different brain areas.

Given the prior literature showing effects of *Cntnap2* loss on synapses in other brain regions, we were surprised that striatal synapses were not impacted in our model. We agree with the reviewer that the variable changes in synaptic properties across brain regions in *Cntnap2* mutant mice is likely a result of distinct compensatory changes in these regions. Differences may also arise depending on whether the synaptic changes originate from the post-synaptic cell or from pre-synaptic changes. An interesting direction for future studies would be to explore the developmental trajectory of excitability and synaptic changes to determine which may be initial perturbations versus those that are secondary and potentially compensatory.

Line 138: "synaptic excitability". How is this term defined? Consider "synaptic changes" instead.

“Synaptic excitability” was used to mean a change in the number and/or function of glutamate receptors. We have now changed this term to “excitatory synaptic changes.”

Consider a short paragraph to highlight some limitations of this study. For example, it is unclear whether SPN hyperexcitability results from a compensatory change in Cntnap2 KO mice and whether the behavioral phenotype is solely due to this cellular phenotype. The study focuses on cortical projections onto SPNs, but these cells receive inputs from other brain areas that were not explored. Lastly, no clear explanation emerges for the variable phenotype in different brain areas and cell types.

We thank the reviewer for this suggestion and have added several paragraphs to the discussion highlighting some limitations of this study.

We hypothesize that the dSPN hyperexcitability in *Cntnap2^-/-^* mice is a primary change, due to the direct relationship between Caspr2 and Kv1.2 channels. The results of our a-DTX experiments suggest that the function and/or contribution of these channels to excitability is altered in *Cntnap2^-/-^* dSPNs. However, it is possible that there are additional changes in dSPNs that occur as a result of *Cntnap2* loss and contribute to the hyperexcitability of these cells. Rather surprisingly, we don’t find evidence for altered excitatory (specifically from cortical inputs) or inhibitory synaptic function, suggesting lack of engagement of homeostatic mechanisms at the synaptic level.

We have not yet determined whether there is a causative relationship between the change in dSPN excitability and the behavioral alterations in *Cntnap2^-/-^* mice. This is a limitation of the current study. In our discussion section, we highlight that the change in dSPN excitability we observe in dorsolateral striatum (DLS) is sufficient to enhance rotarod learning in other mouse models and thus supports a connection between this cellular alteration and behavior. For the other behaviors we measured, we acknowledge that both DLS and other striatal or extra-striatal brain regions have been implicated in these behaviors, and therefore less of a direct connection can be made.

In terms of the inputs, we focused on cortical inputs given their known role in mediating motor and habit learning (PMID: 15242609, 16237445, 19198605). Notably, corticostriatal synapses are altered across a variety of mouse models with mutations in ASD risk genes and therefore may be a point of convergence for disparate genetic insults (PMID: 31758607). We agree that the striatum receives inputs from a variety of brain regions, notably the thalamus, which we did not explore in this study. This would be an interesting area for future studies.

Finally, it is difficult to speculate on the root of the varying impacts to excitability that occur across different brain regions and cell types in *Cntnap2^-/-^* mice. Please see above response to Reviewer #3 for some speculation on this point in regard to the potential involvement of Kv1.2 in the excitability changes in various *Cntnap2^-/-^* cell types. To expand upon this, it is known that ASD-associated mutations can have varying impacts on cell function even across similar cell types within a given brain region – we have seen this between dSPNs and iSPNs (this study, PMIDs: 34380034, 39358043), as have other groups studying ASD risk gene mutations in striatum (PMID: 24995986). This differential impact of the same mutation on intrinsic and/or synaptic physiology across cell types has been identified in other brain regions as well (PMID: 22884327, 26601124). Differences in transcriptional programs, protein expression, neuronal morphology, synaptic inputs and plasticity state make up a non-exhaustive set of variables that will impact the physiological function of a neuron, both in terms of the direct but also indirect consequences of an ASD risk gene mutation. To better address this important question, future studies would benefit from a systematic approach to assessing physiological changes in a given ASD mouse model, both across development and across brain regions.